# Fast yet Safe: Early-Exiting with Risk Control

**Metod Jazbec**[1,*]   **Alexander Timans**[1,*]   **Tin Hadži Veljković**[1]
**Kaspar Sakmann**[2]   **Dan Zhang**[2]   **Christian A. Naesseth**[1]   **Eric Nalisnick**[1,3]
[1]UvA-Bosch Delta Lab, University of Amsterdam
[2]Bosch Center for AI, Robert Bosch GmbH   [3]Johns Hopkins University

## Abstract

Scaling machine learning models significantly improves their performance. However, such gains come at the cost of inference being slow and resource-intensive. Early-exit neural networks (EENNs) offer a promising solution: they accelerate inference by allowing intermediate layers to 'exit' and produce a prediction early. Yet a fundamental issue with EENNs is how to determine *when* to exit without severely degrading performance. In other words, when is it 'safe' for an EENN to go 'fast'? To address this issue, we investigate how to adapt frameworks of *risk control* to EENNs. Risk control offers a distribution-free, *post-hoc* solution that tunes the EENN's exiting mechanism so that exits only occur when the output is of sufficient quality. We empirically validate our insights on a range of vision and language tasks, demonstrating that risk control can produce substantial computational savings, all the while preserving user-specified performance goals.

## 1   Introduction

As predictive models continue to grow in size, so do the costs of running them at inference time [12, 80]. This presents a challenge to domains ranging from mobile computing to smart appliances to autonomous vehicles – all of which require models that operate on resource-constrained hardware [62, 50, 47]. Even if computation is not limited by hardware, concerns over the energy usage and carbon footprint of large models motivates their efficient implementation [55, 44]. Additionally, since computational constraints can be dynamic, *e.g.*, due to variable loads in web traffic or energy demand, it is desirable for models to be able to adjust their computational needs to changing conditions [63].

*Early-exit neural networks* (EENNs) present a simple yet effective approach to such dynamic computation [70, 36]. Leveraging the neural network's compositional nature, EENNs can generate predictions at intermediate layers, thereby 'exiting' the computation 'early' when a stop condition is met. This early-exit ability has proven useful in settings ranging from vision and language to recommendations ([29], see § 4). Yet the flexibility of EENNs does not come for free: predictions generated at early exits are usually inferior to those produced by the full model. In turn, a dilemma arises in which the exit condition must balance computational savings with predictive performance.

In this work, we address the EENN's efficiency *vs.* performance trade-off via statistical frameworks of *risk control* (RC) [4, 8]. By tuning the EENN's exiting mechanism based on a user-specified notion of risk, RC aims to enhance the safety of early-exit outputs. We consider several risks that quantify the difference between the early-exit and full model's outputs, both in terms of prediction quality and uncertainty estimation. Moreover, we study RC frameworks that control the risk with varying degrees of stringency (*i.e.*, in expectation *vs.* with high probability). We demonstrate the effectiveness of this light-weight, *post-hoc* solution across a range of tasks, including image classification, semantic segmentation, language modeling, image generation with diffusion, and speculative decoding in large language models. In particular, we make the following contributions:

---

*Equal contributions. Corresponding authors: <`m.jazbec@uva.nl`, `a.r.timans@uva.nl`>

38th Conference on Neural Information Processing Systems (NeurIPS 2024).

- We formalize EENNs as risk-controlling predictors, ensuring risk control is amenable to the early-exit setting by explicitly linking risk control and early-exit requirements (Prop. 1 & Prop. 2).

- We propose risk functions to control early-exit performance both in terms of model predictions and their underlying predictive distributions (Eq. 6 & Eq. 8). Previous work has considered only prediction quality [65], not uncertainty quality (as we do).

- We improve upon prior work for language modeling [65], demonstrating that our adaptions of risk control allow for less conservative early-exiting and result in larger efficiency gains (§ 3.3, § 5.3).

- We apply, for the first time, risk control to early-exiting in image classification, semantic segmentation, and image generation, as well as speculative decoding in large language models (§ 5).

## 2 Background

**Data.** Let $\mathcal{X} \times \mathcal{Y}$ denote the sample space and assume a data-generating distribution $\mathcal{P}$ over it. We consider $\mathcal{Y} := \{1, \ldots, K\}$ for classification and $\mathcal{Y} \subseteq \mathbb{R}^d$ for regression. Observed samples from $\mathcal{P}$ are split into disjoint train, calibration and test sets, denoted $\mathcal{D}_{train}$, $\mathcal{D}_{cal}$, and $\mathcal{D}_{test}$. We assume samples $(\boldsymbol{x}, \boldsymbol{y})$ in $\mathcal{D}_{cal}$ and $\mathcal{D}_{test}$ to be drawn *i.i.d.* from $\mathcal{P}$, whereas $\mathcal{D}_{train}$ is permitted to be drawn randomly from a different distribution (of same support).

**Early-Exit Neural Networks.** EENNs extend traditional *static* network models by dynamically adjusting computations during the model's forward pass (*e.g.*, the number of evaluated network layers) on the basis of an input sample's complexity or 'difficulty'. More formally, we define an EENN as a sequence of probabilistic classifiers $\hat{p}(\mathbf{y} \mid \mathbf{x} = \boldsymbol{x}; \boldsymbol{\phi}_l, \boldsymbol{\theta}_l)$, where $l = 1, \ldots, L$ enumerates the model's exit layers, and $\boldsymbol{\phi}_l$ and $\boldsymbol{\theta}_l$ define the model's classification head and backbone parameters at the $l$-th exit, respectively. The final index $L$ denotes the full model, *i.e.*, all layers are evaluated for a given input sample $\boldsymbol{x}$. The obtained predictive distribution $\hat{p}(\mathbf{y} \mid \mathbf{x} = \boldsymbol{x}; \boldsymbol{\phi}_l, \boldsymbol{\theta}_l)$ at the $l$-th exit layer, denoted in short as $\hat{p}_l(\mathbf{y}|\boldsymbol{x})$, permits to retrieve both a predicted class label $\hat{\boldsymbol{y}}_l = \arg\max_{\boldsymbol{y} \in \mathcal{Y}} \hat{p}_l(\boldsymbol{y} \mid \boldsymbol{x})$ and an associated confidence score $\hat{c}_l \in [0, 1]$, which aims to capture model's certainty about the exit's current prediction. One common choice for classification tasks is the maximum class probability $\hat{c}_l = \max_{\boldsymbol{y} \in \mathcal{Y}} \hat{p}_l(\boldsymbol{y} \mid \boldsymbol{x})$. However, different notions of confidence are possible, as we explore in § 5.2. At test-time, these confidence scores can be leveraged to determine the early-exit model's required computations for new samples via thresholding. For a given test input $\boldsymbol{x}$, the EENN exits[2] at the first layer for which its confidence exceeds a pre-specified threshold value $\lambda_l \in [0, 1]$. For simplicity, a single threshold value $\lambda_l = \lambda$, $\forall l \in \{1, \ldots, L-1\}$ is often fixed across exit layers, a setup we also consider here. The predictive distribution obtained from the model's early-exit mechanism is then given by

$$\hat{p}_\lambda(\mathbf{y}|\boldsymbol{x}) := \hat{p}_e(\mathbf{y}|\boldsymbol{x}), \text{ where } e = \begin{cases} \min E & \text{if } E \neq \emptyset \\ L & \text{if } E = \emptyset \end{cases}, \ E := \{l \in \{1, ..., L-1\} : \hat{c}_l \geq \lambda\}. \quad (1)$$

The threshold parameter $\lambda$ regulates the trade-off between the EENN's accuracy and efficiency gains. Lower values equate larger speed-ups by increasing the likelihood of an early exit (and *vice versa*), at the cost of generally inferior predictions. Such *marginally monotone* behavior, where model performance improves *on average* across exits, is a core assumption for the practical use of EENNs (see also Fig. 1). We formalize it as

$$\mathbb{E}_{(\boldsymbol{x},\boldsymbol{y})\sim\mathcal{P}}[\ell(\hat{p}_l(\mathbf{y}|\boldsymbol{x}), \boldsymbol{y})] \geq \mathbb{E}_{(\boldsymbol{x},\boldsymbol{y})\sim\mathcal{P}}[(\ell(\hat{p}_{l+1}(\mathbf{y}|\boldsymbol{x}), \boldsymbol{y})] \quad \forall l = 1, \ldots, L-1 \quad (2)$$

for some arbitrary loss function $\ell$, and elaborate on its connection to risk control in § 3.3. It is common to determine early-exiting criteria by investigating these trade-offs between performance and efficiency on hold-out data, selecting thresholds that ensure the EENN meets a user's computational budget [36, 70] or performance goals [16, 21, 48]. A standard practice is to treat the EENN's predictive confidence (*e.g.*, its softmax scores) as a heuristic for prediction quality. However, this is fallible, as EENNs can exhibit fluctuating or poorly-calibrated confidences [40, 37, 51], motivating more principled threshold selection.

---

[2] Such model-driven exiting is distinct from *anytime* settings, where exits are environment-driven [85].

**Risk Control.** Statistical frameworks for risk control (RC) [1, 4, 8] aim to improve prediction reliability by equipping threshold-based models with safety assurances. Specifically, consider a pre-trained prediction model $\hat{f}\lambda$ whose outputs depend on a threshold $\lambda$. For example, given a classification task, the set predictor $\hat{f}\lambda : \mathcal{X} \to 2^{\mathcal{Y}}$ described by Bates et al. [8] includes a class label in the set if its probability exceeds the threshold, *i.e.*, $\hat{f}_\lambda(\boldsymbol{x}) := \{\boldsymbol{y} \in \mathcal{Y} : \hat{p}(\boldsymbol{y}|\boldsymbol{x}) \geq \lambda\}$. Next, a notion of error for $\hat{f}_\lambda$ is captured by defining a problem-specific loss function $\ell : \mathcal{Y} \times \mathcal{Y} \to \mathbb{R}$. For instance, a meaningful choice for the set predictor could be the miscoverage loss $\ell(\hat{f}_\lambda(\boldsymbol{x}), \boldsymbol{y}) = \mathbb{1}[\boldsymbol{y} \notin \hat{f}_\lambda(\boldsymbol{x})]$, where $\mathbb{1}[\cdot]$ is the indicator function. The risk associated with a particular threshold $\lambda \in \Lambda$ is then defined as the expected loss

$$\mathcal{R}(\lambda) := \mathbb{E}_{(\boldsymbol{x},\boldsymbol{y}) \sim \mathcal{P}}\big[\ell(\hat{f}_\lambda(\boldsymbol{x}), \boldsymbol{y})\big], \tag{3}$$

with $\Lambda$ the set of potential threshold candidates. RC frameworks leverage different probabilistic tools – which we detail further in § 3.3 – to determine a subset $\hat{\Lambda} \subseteq \Lambda$ for which the risk in Eq. 3 is guaranteed to be small. Note that $\hat{\Lambda}$ is retrieved in a *post-hoc* manner by leveraging the calibration set $\mathcal{D}_{cal}$ sampled *i.i.d.* from $\mathcal{P}$. Thus, $\mathcal{R}(\hat{\lambda})$ is a random quantity dependent on $\mathcal{D}_{cal}$ for any $\hat{\lambda} \in \hat{\Lambda}$.

Given such a risk, desired safety assurances may vary in strength. For a tolerated risk level $\epsilon \in (0, 1)$, risk control *in expectation* seeks to guarantee that

$$\mathbb{E}_{\mathcal{D}_{cal} \sim \mathcal{P}^n}\big[\mathcal{R}(\hat{\lambda})\big] \leq \epsilon \quad \forall \hat{\lambda} \in \hat{\Lambda}, \tag{4}$$

where the outer expectation is taken over randomly drawn calibration data of finite size $|\mathcal{D}_{cal}| = n$. A stronger statement on risk control *with high probability* requires additionally specifying a probability level $\delta \in (0, 1)$, and aims to ensure that

$$\mathbb{P}_{\mathcal{D}_{cal} \sim \mathcal{P}^n}\big(\mathcal{R}(\hat{\lambda}) \leq \epsilon\big) \geq 1 - \delta \quad \forall \hat{\lambda} \in \hat{\Lambda}. \tag{5}$$

That is, rather than the average control over calibration data in Eq. 4, risk control according to Eq. 5 holds with high probability for any particular sampled set $\mathcal{D}_{cal}$. In both cases, we may refer to $\hat{f}_{\hat{\lambda}}$ for any $\hat{\lambda} \in \hat{\Lambda}$ as a *risk-controlling predictor*. The risk level $\epsilon$ and probability level $\delta$ are user-specified parameters dictating how tightly the risk is controlled, and a particular choice has to consider the problem-specific setting and loss $\ell$. For example, a reasonable choice for the stated miscoverage loss may be $(\epsilon, \delta) = (0.05, 0.1)$. Observing $\hat{\Lambda} = \emptyset$ implies that there is no risk-controlling predictor for the selected $(\epsilon, \delta)$, indicating overly stringent risk control conditions which $\hat{f}_\lambda$ cannot satisfy.

The prediction guarantees obtained via RC are highly practical, since they are *(i)* distribution-free, *i.e.*, they do not impose any particular assumptions on the generating distribution $\mathcal{P}$, *(ii)* are *post-hoc* applicable to any arbitrary choice of underlying predictor $\hat{f}_\lambda$, and *(iii)* hold in finite samples, thus not relying on asymptotic limit statements. Indeed, we experimentally find that the provided assurances hold even for remarkably small calibration sets ($n \approx 100$, see § 5).

## 3 Safe Early-Exiting via Risk Control

We now detail our approach for early-exiting with safety guarantees based on risk control. We begin by outlining EENNs as risk-controlling predictors (§ 3.1). Next, we describe two general types of risk to measure performance drops resulting from early-exiting. Importantly, these risks can be employed to assess the quality of both predictions and predictive distributions (§ 3.2). Finally, we motivate and formalise how different risk control frameworks can be adapted to the early-exit setting (§ 3.3).

### 3.1 EENNs as Risk-Controlling Predictors

As mentioned in § 2, risk control requires a predictor $\hat{f}_\lambda$ whose outputs depend on a threshold $\lambda \in \Lambda$. The EENN's confidence-based thresholding behaviour (following Eq. 1) lends itself naturally to such a formulation. For a particular exit threshold $\lambda \in [0, 1]$, the EENN $\hat{p}_\lambda(\boldsymbol{y}|\boldsymbol{x})$ will act as such a threshold predictor. To ensure that the EENN satisfies the user's control requirements, the risk-controlling threshold set $\hat{\Lambda}$ needs to be identified. Importantly, this can be done *post-hoc* using a pre-trained EENN with fixed weights, since only the exit threshold $\lambda$ is modified. In order to maximize computational savings while ensuring that the user-defined risk is managed, we select $\hat{\lambda} := \min \hat{\Lambda}$, since a low threshold encourages earlier exiting. If $\hat{\Lambda} = \emptyset$ is empty, we default to $\hat{\lambda} = 1$, the equivalent of relying strictly on the full model output $\hat{p}_L(\mathbf{y}|\boldsymbol{x})$.

## 3.2 Early-Exiting Risks

We next detail two types of risk which can be employed to guard against performance drops due to early-exiting. Similarly to Schuster et al. [65], our risks are defined in terms of relative exit performance, permitting their calculation for both labelled and unlabelled calibration data. Moving beyond their setting, we suggest these risks for controlling the quality of both model *predictions* and *predictive distributions*, from which confidence scores can be derived.

**Performance Gap Risk.** When calibration labels are present, these can be used to measure the early-exit performance through supervised losses. Let $\hat{o}_l(x)$ denote a general EENN output for some input $x$. It takes the form $\hat{y}_l$ for predictions and $\hat{p}_l(y|x)$ for the underlying predictive distribution. The supervised *performance gap risk* is then defined as

$$\mathcal{R}^G(\lambda) := \mathbb{E}_{(x,y)\sim\mathcal{P}}\big[\ell\big(\hat{o}_\lambda(x), y\big) - \ell\big(\hat{o}_L(x), y\big)\big], \tag{6}$$

where $\hat{o}_\lambda(x)$ and $\hat{o}_L(x)$ refer to early-exit and full model outputs, respectively. The choice of loss function $\ell$ is task-specific, and we outline relevant choices in § 5, such as the 0-1 loss for image classification. For predictive distribution control, we suggest leveraging a squared distributional loss which, when averaged across samples, recovers the Brier score [13]. Specifically, we define such a 'Brier loss' for classification tasks as

$$\ell_B(\hat{p}_l(y|x), y) := \sum_{k=1}^{K} \big(\hat{p}_l(k|x) - \mathbb{1}[y = k]\big)^2, \tag{7}$$

where $\hat{p}_l(k|x)$ denotes the predicted probability of a particular class $k$, and $\mathbb{1}[y = k]$ its one-hot encoded label. The Brier score is a *strictly proper scoring rule* [26, 27], ensuring its suitability to assess probabilistic forecasts. Moreover, its mathematical formulation lends itself favorably to risk control when compared to other widely used probabilistic metrics. We defer further details to § A.3. Addressing risk control of the underlying predictive distribution $\hat{p}_l(y|x)$ is a compelling extension, as confidence or uncertainty estimates are typically derived from it. Particularly in safety-critical scenarios where reliable uncertainties are essential [32, 9], such control can thus prove very useful.

**Consistency Risk.** In the case of unlabelled calibration data, an unsupervised version of Eq. 6 can be obtained by replacing the ground truth labels $y$ with labels $\hat{y}_L$ obtained from the full model. We define the unsupervised *consistency risk* as

$$\mathcal{R}^C(\lambda) := \mathbb{E}_{(x,\cdot)\sim\mathcal{P}}\big[\ell\big(\hat{o}_\lambda(x), \hat{y}_L\big) - \ell\big(\hat{o}_L(x), \hat{y}_L\big)\big], \tag{8}$$

where only input samples $x \sim \mathcal{P}$ are required for its evaluation.

For regular prediction losses, Eq. 8 collapses to evaluating the per-sample loss $\ell(\hat{y}_\lambda, \hat{y}_L)$, since $\ell(\hat{y}_L, \hat{y}_L) = 0$. For predictive distribution control, the loss difference remains, and we substitute $y$ in Eq. 7 by sampling a label $\hat{y}_L \sim \hat{p}_L(y|x)$ from the EENN's final layer. Our reliance on the last layer's output is motivated by the EENN's marginal monotonicity (Eq. 2 and Fig. 1). Finally, we note that both the performance gap risk $\mathcal{R}^G(\lambda)$ and consistency risk $\mathcal{R}^C(\lambda)$ are quite agnostic to the EENN's actual predictive performance. In both cases, the risk formulation aims to ensure prediction consistency via the *relative* performance gap between exits, as opposed to *absolute* performance with respect to observed ground truth labels.

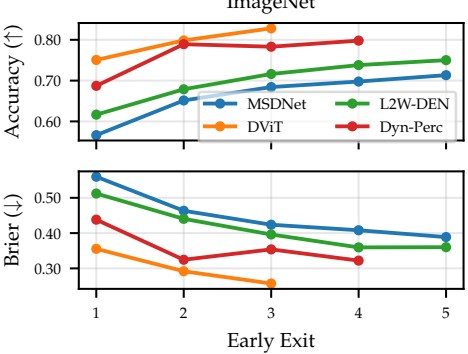

Figure 1: Accuracy and Brier score [13] across exits for different EENNs for image classification on ImageNet (§ 5.1). *Marginally monotone* performance trends (Eq. 2) are generally observed across models, with last-layer exits performing best.

## 3.3 Risk Control Frameworks

After having defined our early-exit risks, we next outline how a desired risk-controlling exit threshold $\hat{\lambda}$ can be computed based on calibration data. We begin by considering a 'naive' empirical approach, followed by risk control *in expectation* (Prop. 1) and *with high probability* (Prop. 2).

**Empirical Approach.** For a tolerated risk level $\epsilon \in (0,1)$, a 'naive' empirical threshold can be selected by picking the smallest threshold $\lambda \in [0,1]$ from the candidate set $\Lambda$ for which the risk on the calibration set $\mathcal{D}_{cal}$ is controlled, *i.e.*,

$$\hat{\lambda}_{\mathrm{emp}} := \min\{\lambda \in \Lambda : \hat{\mathcal{R}}(\lambda; \mathcal{D}_{cal}) \leq \epsilon\}. \tag{9}$$

Note that $\hat{\mathcal{R}}(\lambda; \mathcal{D}_{cal}) = \frac{1}{n} \sum_{i=1}^{n} \ell(\hat{\boldsymbol{o}}_\lambda(\boldsymbol{x}_i), \boldsymbol{y}_i)$ is the *empirical* calibration risk, an approximation of the true risk in Eq. 3 computed on $\mathcal{D}_{cal}$ (and likewise $\hat{\mathcal{R}}(\lambda; \mathcal{D}_{test})$ denotes the empirical test risk). For the risks introduced in § 3.2 the threshold $\hat{\lambda}_{\mathrm{emp}}$ is always well-defined, since $\hat{\mathcal{R}}(\lambda = 1)$ is zero.

**Risk Control in Expectation.** The threshold $\hat{\lambda}_{\mathrm{emp}}$ is a straight-forward choice, but can fail to control the risk on test data if the approximation quality of $\mathcal{R}(\lambda)$ by $\hat{\mathcal{R}}(\lambda; \mathcal{D}_{cal})$ is poor, *e.g.*, due to badly drawn calibration data. Perhaps surprisingly, only a slight modification of Eq. 9 is required to ensure risk control *in expectation*. Specifically, for a bounded loss function $\ell \leq B$ where $B > 0$, and assuming a monotone risk $\mathcal{R}(\lambda)$, the threshold[3]

$$\hat{\lambda}_{\mathrm{CRC}} := \min\left\{\lambda \in \Lambda : \frac{n}{n+1}\hat{\mathcal{R}}(\lambda; \mathcal{D}_{cal}) + \frac{B}{n+1} \leq \epsilon\right\} \tag{10}$$

guarantees Eq. 4, thus shielding against bad sample draws on average. It is easy to show that $\hat{\lambda}_{\mathrm{emp}} = \lim_{n\to\infty} \hat{\lambda}_{\mathrm{CRC}}$, and since our losses are designed to be upper-bounded by $B \in \{1, 2\}$, the two thresholds already coincide for small calibration sets ($n \approx 100$). We formalize risk control in expectation in the following proposition for our early-exit setting:

**Proposition 1.** *Let $\ell : \Lambda \to (-\infty, B]$ be a right-continuous bounded loss, and assume a marginally monotone EENN (Eq. 2). Then the exit threshold $\hat{\lambda}_{CRC}$ ensures risk control in expectation,* i.e., *it holds that* $\mathbb{E}_{\mathcal{D}_{cal} \sim \mathcal{P}^n}[\mathcal{R}(\hat{\lambda}_{CRC})] \leq \epsilon$ *for any $\epsilon \in (0,1)$.*

Our proposition is an extension of *Conformal Risk Control* [4] (CRC) to the early-exit setting, and a proof can be found in § A.1. Our main technical insight is that risk control can be relaxed to assume monotone *risks*, rather than monotone *losses* as in the original formulation [4]. This relaxation is crucial for the early-exit setting, since we can relate monotone risks to assumptions of *marginal* monotonicity on the EENN (see Lemma 1). In contrast, monotone losses translate to assuming *conditional* monotonicity, a much stronger requirement suggesting the EENN's performance improves across exits *per sample*, and which has been shown to be violated in practice [40, 76, 37].

**Risk control with High Probability.** A stronger guarantee can be obtained by ensuring risk control *with high probability* for any drawn calibration set. We employ the *Upper Confidence Bound* (UCB) from Bates et al. [8] for this purpose. First, an empirical upper bound $\hat{\mathcal{R}}^+(\lambda; \mathcal{D}_{cal})$ is derived to bound the risk $\mathcal{R}(\lambda)$ with high probability. That is, for a probability level $\delta \in (0,1)$ it holds that

$$\mathbb{P}_{\mathcal{D}_{cal} \sim \mathcal{P}^n}(\mathcal{R}(\lambda) \leq \hat{\mathcal{R}}^+(\lambda; \mathcal{D}_{cal})) \geq 1 - \delta \quad \forall \lambda \in \Lambda. \tag{11}$$

An exit threshold ensuring risk control according to Eq. 5 is then selected as

$$\hat{\lambda}_{\mathrm{UCB}} := \min\{\lambda \in \Lambda : \hat{\mathcal{R}}^+(\lambda'; \mathcal{D}_{cal}) < \epsilon, \forall \lambda' \geq \lambda\}. \tag{12}$$

Similarly to Prop. 1, we can now formalize risk control with high probability for the early-exit setting:

**Proposition 2.** *Let $\ell : \Lambda \to [-B, B]$ be a bounded loss, and assume a marginally monotone EENN (Eq. 2). Then the exit threshold $\hat{\lambda}_{UCB}$ ensures risk control with high probability,* i.e., *it holds that* $\mathbb{P}_{\mathcal{D}_{cal} \sim \mathcal{P}^n}(\mathcal{R}(\hat{\lambda}_{UCB}) \leq \epsilon) \geq 1 - \delta$ *for any $(\epsilon, \delta) \in (0,1)^2$.*

This reformulation of the main theorem from Bates et al. [8] (Thm. A.1) is proven in § A.1, and an algorithmic description is given in Appendix B. We employ their suggested Waudby-Smith-Ramdas bound [75] (WSR) to compute $\hat{\mathcal{R}}^+(\lambda; \mathcal{D}_{cal})$, but relax the bounding requirements on the loss from $\ell \in [0,1]$ to $\ell \in [-B, B]$ for $B > 0$. This change has important implications for the early-exit setting, since it admits 'rewarding' the EENN when an earlier exit performs better than the final exit for some samples, a phenomenon known as *overthinking* [40, 37]. In practice, this results in a better risk estimate and less conservative early-exiting. See § A.2 for more details on the effect of loss bounds.

---

[3]In Eq. 10 and Eq. 12, we default to $\hat{\lambda} = 1$ if $\hat{\Lambda} = \emptyset$, implying that risk control does not permit early-exiting.

**Learn-then-Test and CALM [65].** *Learn-then-Test* [1] (LTT) is another framework for high-probability risk control, where threshold selection is framed as a multiple hypothesis testing problem. In contrast to UCB (Prop. 2), LTT does not require risk monotonicity, and can thus also be employed when the EENN is suspected to violate marginally monotone behaviour. LTT in the early-exit setting has been employed by Schuster et al. [65] (CALM), presumably motivated by the avoidance of this assumption. However, expecting an EENN to marginally improve across exits is a core requirement which implicitly underlies any practical implementation. Since this assumption is usually also empirically satisfied (Fig. 1), there is no obvious reason to explicitly avoid it. Furthermore, correcting for multiple testing in LTT via *fixed sequence testing* – as is done for CALM – will only yield practical savings if monotonicity is satisfied. We stress these observations since we find that UCB provides greater computational savings than LTT under the same guarantees (Fig. 2 and § 5.3), including

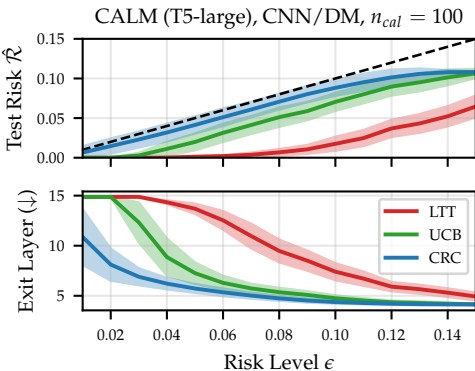

Figure 2: Empirical test risk (*top*) and efficiency gains (*bottom*) for the CALM model [65] for text summarization on CNN/DM. Our adaptation of UCB [8] (Prop. 2) outperforms the LTT [1] approach in CALM by yielding larger efficiency gains under the same risk control assurances (see § 5.3 for details). Shading denotes the standard deviation across $S = 100$ calibration/test splits.

for small-sample regimes ($n \approx 100$), which are of high practical interest and were not explored by Schuster et al. [65]. Moreover, due to LTT's reliance on the Hoeffding-Bentkus bound [11], it cannot account for instances of model overthinking (see § A.2). Thus, unless Eq. 2 is known to be violated, we recommend UCB over LTT in the early-exit setting.

## 4 Related Work

*Early-Exiting* [70, 29] as a dynamic approach to accelerate model inference is both orthogonal and complementary to static model compression techniques such as pruning, quantization, and knowledge-distillation [6, 79, 68, 84, 25]. Its wide-ranging applicability has been demonstrated across numerous vision [36, 48, 15, 69, 22] and language tasks [21, 83, 65, 78, 5, 53]. While most prior work has focused on the trade-off between performance quality and computational savings, the *safety* of early-exit models has received less attention to date [64, 65, 51, 38]. *Risk Control* has gained traction due to an interest in efficient, *post-hoc* approaches with safety assurances for large models. Most related, *conformal prediction* [66] has been popularized as an effective method for uncertainty quantification with guarantees on the miscoverage risk [24, 3]. Recently, multiple proposals address the control of more general risk notions [4, 8, 1, 67, 54, 45], with explored applications ranging from imaging [71, 2, 81, 23, 43, 61, 10, 72] to language [86, 20, 45, 56] and beyond [39, 46]. Most closely related to our work is research by Schuster et al. [65], who first employ risk control for safe early-exiting in language modeling. We move beyond their setting by *(i)* controlling the quality of both model predictions and uncertainty estimates (§ 3.2), *(ii)* obtaining better efficiency gains through careful selection of our risk control framework (§ 3.3), and *(iii)* extending early-exit risk control to novel tasks (§ 5). Ringel et al. [60]'s work is also related: they apply risk control to exit early for a time series prediction task. Yet their emphasis is on exiting from a stream of input features, whereas we exit from the model itself (*i.e.*, a stream of model layers).

## 5 Experiments

We empirically validate early-exiting via risk control on a suite of different tasks: image classification (§ 5.1), semantic segmentation (§ 5.2), language modelling (§ 5.3), image generation with diffusion (§ 5.4), and speculative decoding (§ 5.5). Our code is publicly available at https://github.com/metodj/RC-EENN. We begin by outlining our general risk control design and evaluation metrics.

**Risk control design.** We target control of the performance gap and consistency risks defined in § 3.2. For predictions $\hat{y}_l$ we employ target-specific losses, and, when applicable, for predictive

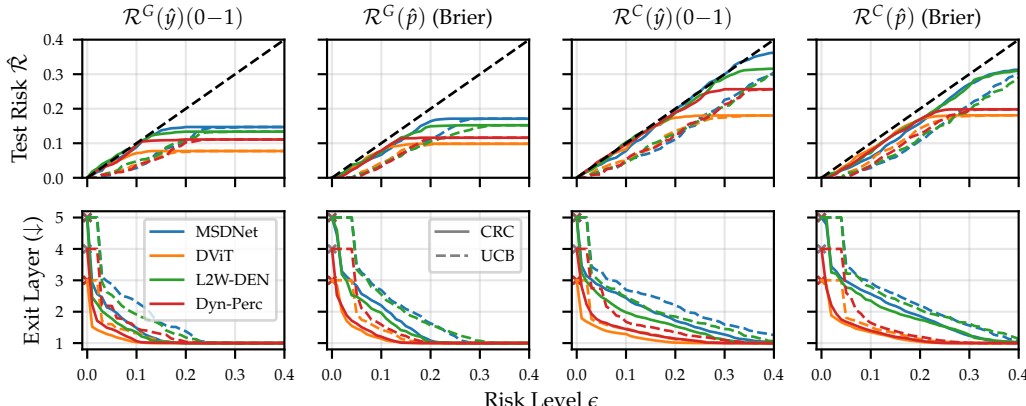

Figure 3: Empirical test risk (*top*) and efficiency gains (*bottom*) for different early-exit models, risks (§ 3.2) and risk levels $\epsilon$ on ImageNet (for calibration set size $n = 100$). In line with theoretical results, the test risk is controlled across models, risk types, and levels. Despite guaranteeing control *in expectation* (CRC, Prop. 1) or *with high probability* (UCB, Prop. 2), obtained gains are substantial.

distributions $\hat{p}_l(\mathbf{y}|\boldsymbol{x})$ our Brier score formulation. We denote these four risks in short as $\mathcal{R}^G(\hat{\boldsymbol{y}})$, $\mathcal{R}^G(\hat{p})$, $\mathcal{R}^C(\hat{\boldsymbol{y}})$ and $\mathcal{R}^C(\hat{p})$. Risk control requirements of different strength are assessed by varying the risk level $\epsilon$. Note that our approach is entirely *post-hoc*, and our experiments leverage existing pretrained EENNs when possible. Thus the underlying models are typically not modified, and we omit the commonly seen performance *vs.* FLOP curves to instead appropriately benchmark against different risk levels. For risk control with high probability, we set $\delta = 0.1$ (*i.e.*, 90 % probability). Reported numbers are averaged across multiple trials of calibration and test splitting ($S = 100$) to account for sampling effects. Additional results across experiments can be found in Appendix D.

**Evaluation metrics.**   We evaluate our results based on obtained test risks and efficiency gains. We assess whether the guarantees stated in § 3.3 are satisfied by checking if the *empirical test risk* for a given risk-controlling threshold is controlled, *i.e.*, $\hat{\mathcal{R}}(\hat{\lambda}; \mathcal{D}_{test}) \le \epsilon$ holds. Ideally, the measured test risk should also approach $\epsilon$ from below, as to prevent overly conservative early-exiting. We measure efficiency gains by reporting the *average exit layer* across test samples, or its relative improvement over last-layer exiting (in %). Similar gains in terms of arithmetic operations (FLOPS) are reported in Appendix D. We favour approaches which, while controlling the test risk, exit as early as possible.

## 5.1   Image Classification

For image classification, we focus on the ImageNet dataset [19]. We employ four state-of-the-art EENNs to demonstrate that our findings generalize across different models and architectures: **MSDNet** [36], **DViT** [74], **L2W-DEN** [30], and **Dyn-Perc** [31] (see § C.1 for details). We employ the standard 0-1 loss for predictions, and the Brier loss formulation from Eq. 7 for predictive distributions. Fig. 3 displays results for the small-sample calibration regime ($n = 100$). In line with our theoretical guarantees, the test risk remains controlled across all models, risk types, and risk levels $\epsilon$ (*top* row). The steeply decreasing efficiency curves affirm that even under strict control requirements, substantial efficiency gains can be obtained (*bottom* row). For example, controlling the prediction gap risk at a strict 5% ($\mathcal{R}^G(\hat{\boldsymbol{y}})$ for $\epsilon = 0.05$) results in a model average of $\sim 61\%$ less layers evaluated for control in expectation (CRC, Prop. 1), and $\sim 46\%$ for control with high probability (UCB, Prop. 2), see Table 2. Naturally, UCB produces more cautious early-exiting due to its stronger safety assurance, but these differences decrease for larger calibration sets (see § D.1 for $n = 1000$). This highlights the practical benefit of allocating more calibration samples: a larger sample size can aid to mitigate the price paid by a high-probability guarantee in terms of obtained inference speed-ups.

## 5.2   Semantic Segmentation

For this task, we explore the effect of different confidence measures used in Eq. 1 on realizable speed-ups. We use the EENN with four exits proposed by Liu et al. [48] (ADP-C, see § C.2). ADP-C permits *pixel-level* early-exiting with per-pixel confidence scores. Since we desire to early-exit the

Table 1: Efficiency gains for semantic segmentation with risk control via UCB (Prop. 2) on Cityscapes. We evaluate for different risks (§ 3.2), confidence measures (§ 5.2) and risk levels $\epsilon$. Displayed values denote relative improvement over last-layer exiting in terms of mean exit layer (in %).

| | Risk | $\mathcal{R}^G(\hat{y})$ (mIoU) | | | $\mathcal{R}^G(\hat{p})$ (Brier) | | | $\mathcal{R}^C(\hat{y})$ (Miscov.) | | | $\mathcal{R}^C(\hat{p})$ (Brier) | | |
|---|---|---|---|---|---|---|---|---|---|---|---|---|---|
| | Level $\epsilon$ | 0.01 | 0.05 | 0.1 | 0.01 | 0.05 | 0.1 | 0.01 | 0.05 | 0.1 | 0.01 | 0.05 | 0.1 |
| Mean | Top-1 | 6.3 | 33.7 | 53.5 | 0.0 | 13.6 | 43.4 | 6.3 | 39.2 | 61.8 | 0.0 | 39.3 | 58.4 |
| | Top-Diff | 9.3 | 35.5 | 54.4 | 0.0 | 17.5 | 44.3 | 6.3 | 39.9 | 62.4 | 0.0 | 38.6 | 57.9 |
| | Entropy | 5.2 | 36.0 | 54.3 | 0.0 | 17.9 | 41.0 | 5.1 | 40.4 | 61.3 | 0.0 | 40.1 | 58.3 |
| Patch | Top-1 | 10.0 | 35.7 | 53.3 | 0.0 | 18.4 | 45.3 | 8.8 | 39.1 | 61.5 | 0.0 | 38.0 | 58.3 |
| | Top-Diff | 10.0 | 35.2 | 53.4 | 0.0 | 19.4 | 45.9 | 8.8 | 40.5 | 62.2 | 0.0 | 38.4 | 58.8 |
| | Entropy | 9.1 | 34.8 | 53.5 | 0.0 | 18.0 | 45.8 | 8.1 | 38.9 | 61.5 | 0.0 | 37.3 | 57.1 |

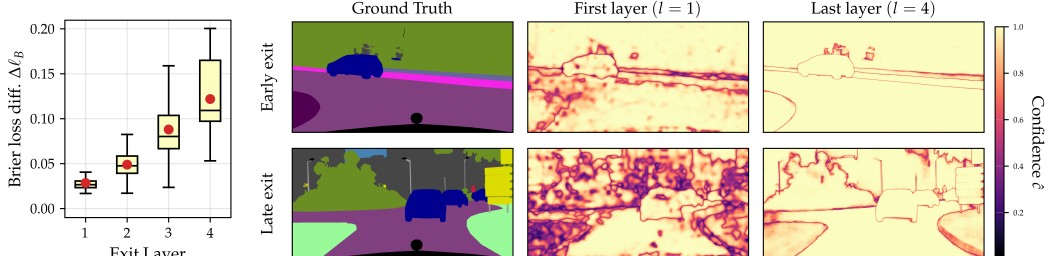

Figure 4: *Right:* Example of our method's early-exiting on Cityscapes [17]. For two samples that exit early ($l = 1$) and exit late ($l = 4$), we display ground truth segmentation masks and confidence maps at the first and last model layer. *Left:* For every sample, we compute the Brier loss difference $\Delta\ell_B = |\ell_B(\hat{p}_1(\mathbf{y}|\boldsymbol{x}), \boldsymbol{y}) - \ell_B(\hat{p}_4(\mathbf{y}|\boldsymbol{x}), \boldsymbol{y})|$ between first and last model layer (Eq. 7), and stratify values across respective exit layers; the red dot denotes the mean. For both figures, we consider the simplest combination of Top-1 confidence score and mean image-level aggregation (for $\epsilon = 0.08$).

entire image instead, we explore *image-level* aggregations alongside different confidence scores, which are briefly outlined below. As task-specific prediction losses, we consider the commonly used *mean intersection-over-union* (mIoU) and *miscoverage* for the labelled and unlabelled cases, respectively. For predictive distribution control, we employ pixel-averaged versions of the Brier loss in Eq. 7 (see § A.3). We evaluate our approaches on Cityscapes validation data (80% $\mathcal{D}_{cal}$, 20% $\mathcal{D}_{test}$); in addition, we finetune and evaluate ADP-C on a subset of the GTA5 dataset [59] in § D.2.

We consider three pixel-level confidence scores: the top class softmax probability (**Top-1**), the difference between top two class probabilities (**Top-Diff**), and the normalized entropy over a pixel's predictive distribution (**Entropy**). In addition, we consider three image-level aggregation strategies: the image's average pixel confidence (**Mean**), its 0.25-th quantile (**Quantile**), and a patch-based approach (**Patch**), wherein a sliding window of fixed size (*e.g.*, $50 \times 50$ pixels) computes the mean confidence over pixels per patch, and the min over such patch scores is retrieved. These aggregations consider both varying levels of prudence (Mean *vs.* Quantile) and granularity (Mean *vs.* Patch).

Table 1 displays obtained efficiency gains for risk control via UCB (Prop. 2) across different risk levels $\epsilon \in \{0.01, 0.05, 0.1\}$. In line with Fig. 3, increased speed-ups are observed as the risk requirements are relaxed (*i.e.*, $\epsilon$ increases). Notably, for a given $\epsilon$ the gains for Brier-based risks tend to be smaller than for prediction risks, affirming more challenging risk control. The differences between combinations of per-pixel confidence and image-level aggregation are most pronounced for small $\epsilon$, where Patch records highest gains while Quantile is more conservative (see § D.2 for full results). In Fig. 4, we display a qualitative example of the model's exiting behaviour. For a sample which exits at the first layer (*top* row), the EENN's confidence map remains fairly stable across subsequent layers, suggesting an accurate model assessment has been reached early on. In contrast, a sample which exits at the final layer (*bottom* row) will see a substantial improvement in model certainty, justifying additional computations. Such behaviour is also visible when stratifying all samples across their respective model exits (Fig. 4, *left*). For samples which exit later, the difference between distributional losses at the first and final layer increases, affirming that compute is spent meaningfully.

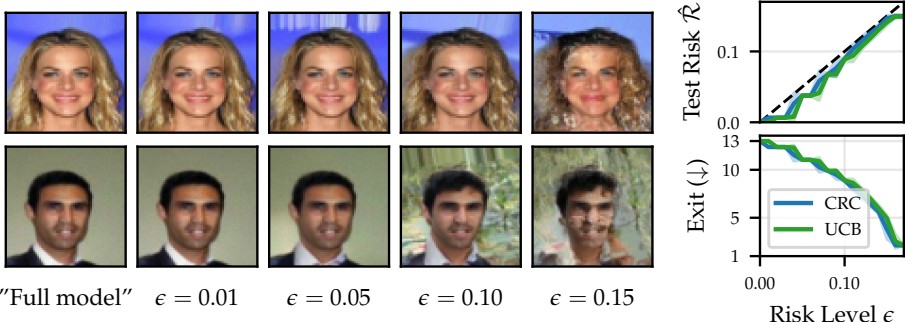

Figure 5: Results for early-exit diffusion with DeeDiff [69] on CelebA [49]. *Left:* The quality of generated images is directly related to the target risk control level $\epsilon$. *Right:* Empirical test risks are controlled for both CRC (Prop. 1) and UCB (Prop. 2) (for calibration set size $n = 500$).

## 5.3 Language Modeling

For this task, we replicate the main experiments from Schuster et al. [65] (CALM), using their early-exit version of the T5 model [57] for text summarization on CNN/DM [33] and question answering on SQuAD [58]. Recall that CALM makes use of the *Learn-then-Test* [1] (LTT) framework for early-exit prediction control, whereas we suggest the *Upper Confidence Bound* [8] (UCB). In contrast to their experiments which involve excessively large calibration sets ($n \approx 8000$), we explore more practical settings of low calibration sample counts with $n \in \{100, 1000\}$. Our results for the *performance gap risk* (Eq. 6) based on task-specific losses (*ROUGE-L* for CNN/DM and *Token-F1* for SQuAD) are displayed in Fig. 2 and § D.3. In all cases, UCB exit thresholds provide larger computational savings over LTT, while ensuring the same risk control with high probability (Eq. 5). Since particularly pronounced for $n = 100$, these results highlight the need for careful framework selection in order to minimize the cost of providing guarantees. Once more, risk control in expectation (CRC, Prop. 1) permits faster exiting due to its weaker safety assurance. Encouragingly, even with as few as $n = 100$ calibration samples, CRC exit thresholds reach near-optimal exiting, as indicated by their proximity to the ideal (diagonal) risk line. This suggests that even for modern language tasks, equipping an EENN with notions of safety does not necessitate a strong compromise on inference efficiency.

## 5.4 Image Generation with Early-Exit Diffusion

To demonstrate the wide-ranging applicability of our proposal, we lastly consider early-exiting for image generation with diffusion. We employ the DeeDiff model [69], which performs early-exiting on the denoising network at each sampling step during the reverse diffusion process[4]. We target control of the *perceptual difference* between images generated by the accelerated and full diffusion processes, which we measure with the LPIPS score [82], and where lower values indicate perceptually closer images. Our results on the CelebA dataset [49] are shown in Fig. 5 for both risk control via CRC (Prop. 1) and UCB (Prop. 2), asserting that the risk is controlled at all levels $\epsilon$. The impact of the risk level on image generation is additionally visualized for two examples. For strict control requirements the early-exit generations perceptually resemble the full model, whereas generated image quality visibly deteriorates for larger $\epsilon$ (but remains controlled). For smaller $\epsilon$, the speed-ups within each sampling step are relatively modest (*e.g.*, ~15% for $\epsilon = 0.05$). However, such gains accrue over the large number of sampling steps in the image generation process (~500), resulting in overall meaningful savings. Similar observations for CIFAR [42] are reported in § D.4.

## 5.5 Speculative Decoding for Large Language Models

While we primarily focus on early-exiting, our final experiment highlights how risk control can also be applied to other techniques for efficient inference. Here we consider accelerating the inference of large language models (LLMs) using the (soft) speculative decoding approach *BiLD* [41]. BiLD uses a small draft model to generate multiple tokens autoregressively while the original LLM is only employed for verification.

---

[4]Since the code for DeeDiff is not publicly available, we implement it ourselves (see § C.4).

This step can be performed in a single forward pass for all proposed tokens, necessitating less computations from the larger, more expensive model. During verification the difference in token distributions between the models is computed, and the tokens generated by the draft model are rejected if the difference exceeds a tolerated threshold, triggering a "rollback" (see Eq. 3 in [41]). We apply our risk control frameworks to this rollback threshold, which dictates *(i)* the similarity of generated text to the output produced solely by the original LLM, and *(ii)* the associated inference speed-ups in terms of sentences (or samples) per second. Our results in Fig. 6 for the *performance gap risk* (Eq. 6), as defined via the difference in sentence-level *BLEU* scores, corroborate our previous findings for language modeling (Fig. 2 and § 5.3). That is, our approaches via CRC (Prop. 1) and UCB (Prop. 2) provide meaningful speed-ups and improve upon the LTT method [1] used by CALM [65], while maintaining the desired risk control across test samples.

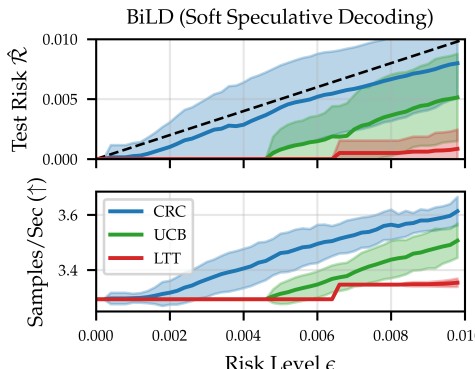

Figure 6: Empirical test risk (*top*) and efficiency gains (*bottom*) for the BiLD model [41] for a machine translation task (De–En) on IWSLT [14] (with $n = 500$). We fix the fallback threshold to $0.5$, and apply risk control to the *rollback* threshold. Our adaptation of UCB again outperforms LTT by yielding larger efficiency gains under the same guarantees. Shading denotes the standard deviation across $S = 100$ calibration/test splits.

## 6   Discussion

Our work addresses how to select a 'safe' exiting mechanism for early-exit neural networks (EENNs). We propose balancing the EENN's efficiency *vs.* performance trade-off via risk control, ensuring that accelerated inference does not compromise the quality of the early-exit outputs. We validate our light-weight, *post-hoc* solution on a variety of tasks and improve upon prior work [65] (see § 4).

**Limitations and Future Work.**   A key limitation of our work is the reliance on a single shared exit threshold among layers (Eq. 1). While using a shared exit threshold is common [69, 48, 40, 76, 83, 77], relaxing this condition could lead to further efficiency gains. This, however, introduces new challenges both in terms of theory (*e.g.*, defining monotonicity requirements) and practice (*e.g.*, substantially larger search spaces). Overcoming them by adopting recently proposed risk control techniques for high-dimensional thresholds [60, 71] could prove promising for future extensions. Another (simple) workaround is to reduce the multi-dimensional problem back to a single threshold by use of a *threshold function*, as partially explored in [65]. Instead of working directly with multiple thresholds, our risk control framework can then be applied to this scalar parameter. Additionally, multiple risk control extensions provide natural avenues for future work. Firstly, risk control as used in our work is achieved only *marginally* across observations (Eq. 3), and one could aspire for more granular *exit-conditional* control [60]. Secondly, the employed risk control frameworks define risk in terms of the *expected* loss. One could instead aim to control the tails of the loss, *e.g.*, via specific quantiles of interest [67]. Lastly, relaxing the *i.i.d* assumption on calibration and test data could help extend risk-controlling EENNs to scenarios with test-time distribution shifts [86] or to online updating strategies [23].

**Broader Impacts.**   EENNs provide a simple and effective approach to dynamic computation [29]. Their computational savings can reduce energy costs and the carbon footprint, as well as allow the model to be deployed on resource-constrained hardware. By incorporating a 'safe' exit mechanism into these models, we improve their trustworthiness and strengthen the reliability of their intermediate outputs, along with any decisions based on them. This facilitates safer model deployment in real-world applications and contributes to more responsible decision-making. While we do not foresee any direct negative consequences from our work, improper use of our risk control framework can lead to violations or misinterpretations of its provided guarantees. This, in turn, can risk instilling a false sense of security. Overall, we believe that our work outlines an effective approach to improve the reliability of EENNs and to safely balance their inherent efficiency *vs.* performance trade-off. In doing so, it contributes to the goal of developing models that are ultimately *fast yet safe*.

## Acknowledgments and Disclosure of Funding

We thank Mona Schirmer, Rajeev Verma, Christoph-Nikolas Straehle, Patrick Forré and Stephen Bates for helpful discussions and clarifications. We are also grateful to the anonymous reviewers who helped improve the work with their constructive feedback. This project was generously supported by the Bosch Center for Artificial Intelligence. Eric Nalisnick did not utilize resources from Johns Hopkins University for this project.

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

# APPENDIX

The appendix is organized as follows:

- Appendix A contains mathematical details, namely proofs for our propositions (§ A.1), elaboration on bounding conditions of the loss function (§ A.2), and further details on our Brier score formulation (§ A.3).

- Appendix B contains algorithmic descriptions of risk control via the *Upper Confidence Bound* [8] (UCB, Prop. 2) with the Waudby-Smith-Ramdas bound [75] (WSR, Prop. 3) in Algorithm 1 and Algorithm 2.

- Appendix C contains various implementation details of our four experiments: image classification (§ C.1), semantic segmentation (§ C.2), language modeling (§ C.3), and image generation (§ C.4).

- Appendix D contains additional results across our suite of experiments (in the same order as above), including risk control and efficiency curves for varying risk levels $\epsilon$ (§ D.1, § D.2, § D.3, § D.4), as well as additional efficiency gain tables (§ D.1, § D.2).

## A  Mathematical Details

### A.1  Proofs for Risk Control

We begin by formalizing the connection between marginal monotonicity requirements on the EENN (Eq. 2) and the monotonicity of risks (Eq. 3) in Lemma 1 below.

**Lemma 1.** *A marginally monotone EENN satisfying Eq. 2 for some arbitrary loss function $\ell$ implies monotone decreasing risks of the form in § 3.2,* i.e., *we have that $\mathcal{R}(\lambda_1) \geq \mathcal{R}(\lambda_2)$ for $\lambda_1 \leq \lambda_2$.*

*Proof.* For a given test sample $\boldsymbol{x}$, denote the exit layers corresponding to exit thresholds $\lambda_1$ and $\lambda_2$ as $l_1$ and $l_2$. From the EENN's confidence-based exiting mechanism in Eq. 1 it follows that $l_1 \leq l_2$, *i.e.*, a smaller exit threshold $\lambda_1$ will result in exits that are earlier or equal to the larger threshold $\lambda_2$. From our risk formulation in terms of relative exit performances in § 3.2 and marginal monotonicity according to Eq. 2 it then follows that

$$\mathcal{R}(\lambda_1) = \mathbb{E}_{(\boldsymbol{x},\boldsymbol{y})\sim\mathcal{P}}[\ell(\hat{\boldsymbol{o}}_{l_1}(\boldsymbol{x}), \boldsymbol{y}) - \ell(\hat{\boldsymbol{o}}_L(\boldsymbol{x}), \boldsymbol{y})]$$
$$\overset{\text{(Eq. 2)}}{\geq} \mathbb{E}_{(\boldsymbol{x},\boldsymbol{y})\sim\mathcal{P}}[\ell(\hat{\boldsymbol{o}}_{l_2}(\boldsymbol{x}), \boldsymbol{y}) - \ell(\hat{\boldsymbol{o}}_L(\boldsymbol{x}), \boldsymbol{y})] = \mathcal{R}(\lambda_2).$$

$\square$

Next, we prove Prop. 1, our adaptation of the *Conformal Risk Control* [4] (CRC) guarantee on risk control *in expectation* for the early-exit setting. Our proof closely follows the proof for Thm. 1 in Angelopoulos et al. [4], but we relax the condition on monotone *losses* to that on monotone *risks*, which implies assuming marginal monotonicity on the EENN according to Lemma 1. We restate our proposition from the main paper for completeness first.

**Proposition 1.** *Let $\ell : \Lambda \to (-\infty, B]$ be a right-continuous bounded loss, and assume a marginally monotone EENN (Eq. 2). Then the exit threshold $\hat{\lambda}_{CRC}$ ensures risk control in expectation,* i.e., *it holds that $\mathbb{E}_{\mathcal{D}_{cal}\sim\mathcal{P}^n}\big[\mathcal{R}(\hat{\lambda}_{CRC})\big] \leq \epsilon$ for any $\epsilon \in (0, 1)$.*

*Proof.* Consider the calibration set $\mathcal{D}_{cal} = \{(\boldsymbol{x}_i, \boldsymbol{y}_i)\}_{i=1}^n \sim \mathcal{P}^n$ and some test sample $(\boldsymbol{x}, \boldsymbol{y}) \sim \mathcal{P}$ drawn *i.i.d* from $\mathcal{P}$, and denote their union set as $\tilde{\mathcal{D}} := \mathcal{D}_{cal} \cup (\boldsymbol{x}, \boldsymbol{y})$. Additionally, define $\ell_i(\lambda) := \ell(\hat{\boldsymbol{o}}_\lambda(\boldsymbol{x}_i), \boldsymbol{y}_i)$ as the loss of the EENN's early-exit output for the $i$-th sample. In particular, $\ell_{n+1}(\lambda)$ now denotes the test sample's loss. Let us first recall the definition of $\hat{\lambda}_{CRC}$ (Eq. 10):

$$\hat{\lambda}_{CRC} := \min\{\lambda \in \Lambda : \frac{n}{n+1}\hat{\mathcal{R}}(\lambda; \mathcal{D}_{cal}) + \frac{B}{n+1} \leq \epsilon\}.$$

Note that $\Lambda$ is a discrete grid of values over $[0, 1]$, *e.g.*, equidistant values $\{0.01, 0.02, \ldots, 1\}$, and ff $\hat{\Lambda} = \emptyset$ and the risk control condition is never met, we default to $\hat{\lambda} = 1$ for all threshold selection

procedures. Thus, the min is well-defined. Next, consider the empirical risk $\hat{\mathcal{R}}_{n+1}(\lambda; \tilde{\mathcal{D}})$ computed over $\tilde{\mathcal{D}}$ using the $n+1$ available samples, and define the following threshold choice:

$$\hat{\lambda}_{n+1} := \min\{\lambda \in \Lambda : \hat{\mathcal{R}}_{n+1}(\lambda; \tilde{\mathcal{D}}) \leq \epsilon\}. \tag{13}$$

$\hat{\lambda}_{n+1}$ is always well-defined, since $\hat{\mathcal{R}}_{n+1}(\lambda = 1)$ is zero for the risks introduced in § 3.2. As we assume a bounded loss function $\ell \leq B$, we observe that for any $\lambda \in \Lambda$ we have

$$\hat{\mathcal{R}}_{n+1}(\lambda; \tilde{\mathcal{D}}) = \frac{n}{n+1}\hat{\mathcal{R}}(\lambda; \mathcal{D}_{cal}) + \frac{\ell_{n+1}(\lambda)}{n+1} \leq \frac{n}{n+1}\hat{\mathcal{R}}(\lambda; \mathcal{D}_{cal}) + \frac{B}{n+1},$$

which implies that $\hat{\lambda}_{n+1} \leq \hat{\lambda}_{\mathrm{CRC}}$. Since we assume a marginally monotone EENN, by Lemma 1 it follows that the risk is monotone decreasing and by $\mathcal{R}(\hat{\lambda}_{n+1}) \geq \mathcal{R}(\hat{\lambda}_{\mathrm{CRC}})$ we also have that

$$\mathbb{E}_{\mathcal{D}_{cal} \sim \mathcal{P}^n}[\mathcal{R}(\hat{\lambda}_{n+1})] \geq \mathbb{E}_{\mathcal{D}_{cal} \sim \mathcal{P}^n}[\mathcal{R}(\hat{\lambda}_{\mathrm{CRC}})]. \tag{14}$$

By Eq. 3 and our loss definition above, we can rewrite for a general $\lambda$ the expression

$$\mathbb{E}_{\mathcal{D}_{cal} \sim \mathcal{P}^n}[\mathcal{R}(\lambda)] = \mathbb{E}_{\mathcal{D}_{cal} \sim \mathcal{P}^n}[\mathbb{E}_{(\boldsymbol{x},\boldsymbol{y}) \sim \mathcal{P}}[\ell_{n+1}(\lambda)]] \overset{i.i.d}{=} \mathbb{E}_{\tilde{\mathcal{D}} \sim \mathcal{P}^{n+1}}[\ell_{n+1}(\lambda)],$$

in short $\mathbb{E}[\ell_{n+1}(\lambda)]$. The remainder of the proof follows Thm. 1 in Angelopoulos et al. [4]. Assume a particular set $\tilde{\mathcal{D}}$ is given. Then the threshold $\hat{\lambda}_{n+1}$ is void of randomness and a constant, and by the *i.i.d* condition we also have that $\ell_{n+1}(\lambda)|\tilde{\mathcal{D}} \sim \mathrm{Unif}(\{\ell_1, \ldots, \ell_{n+1}\})$ is uniformly distributed. Combining these observations with the law of total expectation (l.o.t.e.) and right-continuity (r.c.) of the loss, the final result follows:

$$\mathbb{E}[\ell_{n+1}(\hat{\lambda}_{\mathrm{CRC}})] \overset{(\text{Eq. }14)}{\leq} \mathbb{E}[\ell_{n+1}(\hat{\lambda}_{n+1})] \overset{\text{l.o.t.e.}}{=} \mathbb{E}\big[\mathbb{E}[\ell_{n+1}(\hat{\lambda}_{n+1}) \,|\, \tilde{\mathcal{D}}]\big]$$

$$\overset{\text{Unif}}{=} \mathbb{E}\left[\frac{1}{n+1}\sum_{i=1}^{n+1}\ell_i(\hat{\lambda}_{n+1})\right] \overset{(\text{Eq. }13)\ \&\ \text{r.c.}}{\leq} \mathbb{E}[\epsilon] = \epsilon.$$

$\square$

Next, we sketch a proof for Prop. 2, our adaptation of the *Upper Confidence Bound* [8] (UCB) guarantee on risk control *with high probability* for the early-exit setting. Our main change includes modifying the Waudby-Smith-Ramdas bound [75] (WSR) to relax the bounding condition on the loss from $\ell \in [0,1]$ to $\ell \in [-B, B]$ (Prop. 3). We first restate our proposition from the main paper.

**Proposition 2.** *Let $\ell : \Lambda \to [-B, B]$ be a bounded loss, and assume a marginally monotone EENN (Eq. 2). Then the exit threshold $\hat{\lambda}_{UCB}$ ensures risk control with high probability,* i.e.*, it holds that $\mathbb{P}_{\mathcal{D}_{cal} \sim \mathcal{P}^n}(\mathcal{R}(\hat{\lambda}_{UCB}) \leq \epsilon) \geq 1 - \delta$ for any $(\epsilon, \delta) \in (0,1)^2$.*

*Proof.* Our result follows almost directly from the proofs in Bates et al. [8] (for their Thm. A.1), where we leverage the required risk monotonicity by Lemma 1. We omit the technical requirement on risk continuity from the original proof, since a relaxation to non-continuous risks is permitted ([8], Remark 3). A proof that the WSR bound can be used to construct a valid upper confidence bound can be found in Bates et al. [8], Sec. 3.1.3. However, an assumption on losses bounded to $\ell \in [0,1]$ is made, which is overly restrictive for the early-exit setting. We relax this assumption to $\ell \in [-B, B]$ in Prop. 3 below, concluding the result. $\square$

Since our risk definitions in § 3.2 can naturally assume negative values, and we thus want to account for the occurrence of earlier exits performing better, we relax the bounding condition on the loss function for the Waudby-Smith-Ramdas bound [75] (WSR) in the following proposition.

**Proposition 3.** *A valid upper confidence bound (Eq. 11) based on the Waudby-Smith-Ramdas bound can be constructed for losses $\ell \in [-B, B]$ with $B > 0$.*

*Proof.* Observe the definitions of individual components $(\mu_i, \sigma_i^2, \nu_i, \kappa_i)$ for the WSR bound in Algorithm 2. In particular, define $\nu_i = \min\{1/2B, \sqrt{\frac{2\log(1/\delta)}{n\sigma_{i-1}^2}}\}$. Since the second term is always non-negative, it follows that $\nu_i \in [0, 1/2B]$. For the loss $\ell_i \in [-B, B]$, we then have that $(\ell_i - \mathbb{E}[\ell_i]) \in [-2B, 2B]$. Hence, it follows that $1 - \nu_i(\ell_i - \mathbb{E}[\ell_i]) \geq 0$, which implies that $\kappa_i = \prod_{j=1}^{i}\{1 - \nu_j(\ell_j - \mathbb{E}[\ell_j])\}$ is a non-negative martingale. The rest of the proof follows Prop. 5 from Bates et al. [8], concluding the result. $\square$

Observe that the proof for Prop. 3 utilizes the fraction $1/2B$ in its definition of $\nu_i$ which can make the WSR bound less tight. However, in the case of a marginally monotone EENN (Lemma 1) we can relax this fraction to $1/B$ instead, a comment we formalize in the following remark.

**Remark 1.** *Note that in the case of a marginally monotone EENN, the bound from Prop. 3 can be further optimized. For a bounded loss $\ell \in [0, B]$, the relative early-exit loss $\ell(\lambda, L) := \ell(\hat{\boldsymbol{o}}_\lambda(\boldsymbol{x}), \boldsymbol{y}) - \ell(\hat{\boldsymbol{o}}_L(\boldsymbol{x}), \boldsymbol{y})$ will fall in the $[-B, B]$ range (see also § 3.2 and § A.2). Additionally, due to the marginal monotonicity assumption (Eq. 2) the risk based on the relative loss will be non-negative, i.e., $\mathcal{R}(\lambda, L) = \mathbb{E}[\ell(\lambda, L)] \in [0, B]$. This implies that $(\ell(\lambda, L) - \mathcal{R}(\lambda, L)) \in [-2B, B]$. Hence, $\kappa_i$ will be a non-negative martingale even when the upper bound $1/B$ is used for $\nu_i$ instead of $1/2B$.*

## A.2 On Bounding of the Loss Function

The risks outlined in § 3.2 rely on an early-exit loss definition in terms of relative exit performance. That is, our risks take the general form

$$\mathcal{R}(\lambda) = \mathbb{E}_{(\boldsymbol{x}, \boldsymbol{y}) \sim \mathcal{P}}\big[\ell(\lambda, L)\big], \quad \ell(\lambda, L) := \ell\big(\hat{\boldsymbol{o}}_\lambda(\boldsymbol{x}), \boldsymbol{y}\big) - \ell\big(\hat{\boldsymbol{o}}_L(\boldsymbol{x}), \boldsymbol{y}\big), \tag{15}$$

with $\ell(\lambda, L)$ denoting the relative early-exit loss. Recall that $\hat{\boldsymbol{o}}_\lambda$ and $\hat{\boldsymbol{o}}_L$ are based on $\hat{p}_\lambda$ and $\hat{p}_L$, respectively. For a bounded loss $\ell \in [0, B]$, we then have that $\ell(\lambda, L) \in [-B, B]$. In our early-exit setting, negative losses have an intuitive interpretation. The associated test samples are cases where the EENN *overthinks* [40, 37], *i.e.*, the early-exit $\hat{p}_\lambda$ performs better than the final exit $\hat{p}_L$.

Risk control via CRC (Prop. 1) or UCB (Prop. 2) with the relaxed WSR bound (Prop. 3) conveniently account for such occurrences. In contrast, this presents a challenge for the *Learn-then-Test* [1] (LTT) framework employed by Schuster et al. [65], since the underlying Hoeffding-Bentkus [11] (HB) bound requires $\ell(\lambda, L) \in [0, 1]$. As a workaround, Schuster et al. [65] instead impose a lower loss limit of zero, *i.e.*, they use

$$\ell(\lambda, L) := \max\{0, \ell\big(\hat{\boldsymbol{o}}_\lambda(\boldsymbol{x}), \boldsymbol{y}\big) - \ell\big(\hat{\boldsymbol{o}}_L(\boldsymbol{x}), \boldsymbol{y}\big)\}. \tag{16}$$

While solving their technical requirement, it introduces a key drawback in that the risk control procedure cannot account for samples where the risk requirement is satisfied 'for free'. This introduces substantially more conservative early-exiting (see Fig. 7), since an upper bound on the empirical calibration risk is used for threshold tuning.

In addition, observe that the Brier score is naturally bounded by $[0, 2]$ for the multi-class setting [13]. Thus, our relative early-exit *Brier losses* assume values in the range of $[-2, 2]$. While acceptable for risk control via CRC and UCB (by setting $B = 2$), it once again does not align with the LTT requirement on $\ell(\lambda, L) \in [0, 1]$. Thus, applying LTT requires additional restrictions such as scaling (*e.g.*, with a $1/2$ term) and Eq. 16 to satisfy the bounds. This highlights another drawback where the intuitive risk interpretation as a Brier score difference is partially lost (see § A.3 below).

Note that while CRC and UCB are amenable to $\ell(\lambda, L) \in [-B, B]$, it can still be beneficial to ensure non-negative losses (*e.g.*, by Eq. 16) in order to improve the marginal monotonicity of the EENN (Eq. 2). Namely, it can happen that an EENN is *not* marginally monotone for an unrestricted loss (Eq. 15), but *is* for its zero-bounded counterpart (Eq. 16). Hence, such approaches might be useful when there is reason to believe that EENN violates its marginal monotonicity assumption, though in such cases a practitioner might better opt for the pruning of unnecessary model layers instead.

## A.3 Brier score formulation

**Brier score motivation.** The Brier score is a *strictly proper scoring rule* [26, 27], ensuring its suitability to assess probabilistic forecasts. This can be demonstrated by its decomposition structure, which highlights that both calibration and sharpness properties of the forecaster are taken into account [52, 18]. Moreover, its mathematical formulation lends itself favorably to risk control when compared to other widely used probabilistic metrics. These include the expected calibration error (ECE [28]), which requires binning and thus cannot be defined at a per-sample level; the negative log-likelihood, which is less interpretable and unbounded; the ranked probability score (RPS), which does not treat class distances equally; and the continuous ranked probability score (CRPS [34]) or $f$−divergences like the Hellinger distance, which can be overly conservative by aiming to control the (potentially long) tail of the distribution, and require access to the full (ground truth) distribution. While amenable to our overall risk control framework, such probabilistic metrics, or any derived *top-k* uncertainty measures, seem either less practical or less principled.

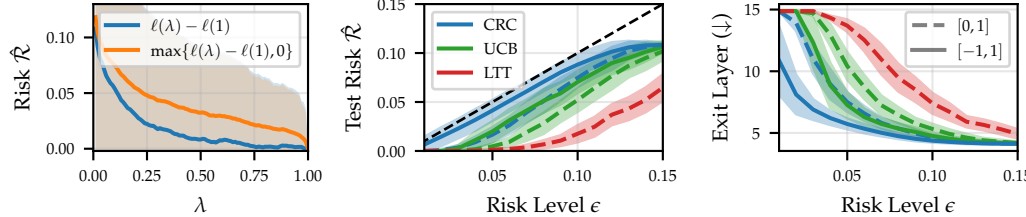

Figure 7: We display risk curves based on relative early-exit losses $\ell(\lambda, L)$ for the CALM model [65] on CNN/DM (*left*). The risk based on the zero-bounded losses (—; Eq. 16) naturally upper-bounds the one without (—; Eq. 15). Since LTT requires $\ell(\lambda, L) \in [0, 1]$, threshold tuning is performed based on the larger risk (—), despite aiming to control the actual risk (—). This results in overly conservative early-exiting, as indicated by the LTT test risk (- -) deviating furthest from the optimal (diagonal) risk line (*middle*), and its lowest empirical gains (*right*). Since CRC and UCB permit negative losses, we display their results for both zero-bounded (- -; Eq. 16) and unrestricted (—; Eq. 15) settings. Also here, the unrestricted setting results in controlled risks with larger efficiency gains, highlighting its relevance for the early-exit setting with relative losses of the form $\ell(\lambda, L)$.

**Brier loss and Brier score.** For convenience (but easily transferable), consider the supervised setting where we target risk control of the *performance gap risk* (Eq. 6) for predictive distributions, which we denote in short as $\mathcal{R}^G(\hat{p})$. This requires computing the *Brier loss* $\ell_B(\hat{p}_l(\mathbf{y}|\boldsymbol{x}), \boldsymbol{y})$, which we define for the multi-class setting in Eq. 7. Now consider a dataset $\mathcal{D} \sim \mathcal{P}$ of size $N$ (*e.g.*, the calibration set $\mathcal{D}_{cal}$). The associated Brier score [13] denoted $\texttt{Brier}(\hat{p}_l)$ for the $l$-th exit layer is then defined as the mean Brier loss across samples, *i.e.*, we have

$$\texttt{Brier}(\hat{p}_l) = \frac{1}{N} \sum_{n=1}^{N} \ell_{B,n}(\hat{p}_l) = \frac{1}{N} \sum_{n=1}^{N} \ell_B(\hat{p}_l(\mathbf{y}|\boldsymbol{x}_n), \boldsymbol{y}_n) \overset{(\text{Eq. 7})}{=} \frac{1}{N} \sum_{n=1}^{N} \sum_{k=1}^{K} \left(\hat{p}_l(k|\boldsymbol{x}_n) - \mathbb{1}[\boldsymbol{y}_n = k]\right)^2,$$

where $\ell_{B,n}(\hat{p}_l)$ is an abbreviation for the $n$-th sample's Brier loss. The risk $\mathcal{R}^G(\hat{p})$ that we aim to control is approximated by its empirical equivalent $\hat{\mathcal{R}}^G(\hat{p})$ on $\mathcal{D}$, and can then be interpreted as the difference in Brier scores between our EENN with threshold $\lambda$ and the full (last-layer exit) model:

$$\hat{\mathcal{R}}^G(\hat{p}) = \frac{1}{N} \sum_{n=1}^{N} \left[\ell_B(\hat{p}_\lambda(\mathbf{y}|\boldsymbol{x}_n), \boldsymbol{y}_n) - \ell_B(\hat{p}_L(\mathbf{y}|\boldsymbol{x}_n), \boldsymbol{y}_n)\right]$$

$$= \frac{1}{N} \sum_{n=1}^{N} \ell_{B,n}(\hat{p}_\lambda) - \frac{1}{N} \sum_{n=1}^{N} \ell_{B,n}(\hat{p}_L) = \texttt{Brier}(\hat{p}_\lambda) - \texttt{Brier}(\hat{p}_L).$$

**Mean-pixel Brier score.** Since our semantic segmentation experiment (§ 5.2) is a pixel-level classification task, we obtain *pixel-level* predictive distributions, and thus compute per-pixel Brier losses. For an image of height $H$ and width $W$, we can denote by $\ell_{B,n}(\hat{p}_{l,i,j})$ the $n$-th sample's Brier loss at the pixel location $(i, j) \in H \times W$. Since we target *image-level* early-exiting, these per-pixel Brier losses are averaged across pixels to compute a per-image Brier loss, which in turn is averaged across samples to obtain the $l$-th exit layer's Brier score. That is, the layer's associated Brier score $\texttt{Brier}(\hat{p}_l)$ is defined as

$$\texttt{Brier}(\hat{p}_l) = \frac{1}{NHW} \sum_{n=1}^{N} \sum_{(i,j) \in H \times W} \ell_{B,n}(\hat{p}_{l,i,j}) = \frac{1}{NHW} \sum_{n=1}^{N} \sum_{(i,j) \in H \times W} \ell_B(\hat{p}_l(\mathbf{y}|\boldsymbol{x}_{n,i,j}), \boldsymbol{y}_{n,i,j}),$$

and can be interpreted as the *mean-pixel* Brier score of the particular exit layer. Note that if we interpret every image pixel as an individual sample (*i.e.*, define $\tilde{N} := NHW$), the Brier score formulation as a sample-averaged Brier loss continues to hold. Similar to above, the targeted risk is then interpreted as the *mean-pixel* Brier score difference.

# B Algorithmic Details

Here, we sketch the algorithm for computing the risk-controlling threshold $\hat{\lambda}_{\text{UCB}}$ (Eq. 12) via the *Upper Confidence Bound* [8] (UCB, Prop. 2) with the Waudby-Smith-Ramdas bound [75] (WSR, Prop. 3). The algorithm is an adaptation of the approach presented in Bates et al. [8] to the early-exit setting. Note that our practical code implementation differs slightly from the pseudo-code presented here, as we omit here some code optimization steps such as vectorization to improve readability.

---

**Algorithm 1:** Risk control for EENNs via UCB (Prop. 2)

---

**input** : EENN $\hat{p}_\lambda, \mathcal{D}_{cal}, \epsilon, \delta,$ `loss function` $\ell,$ `grid step` $\Delta$
**output:** $\hat{\lambda}_{\text{UCB}}$

```
grid = np.arange(1,0,-Δ)
# Construct the UCB (Eq. 11) UCB = np.ones(len(grid))
```
**for** $i, \lambda$ *in grid* **do**
    $\quad$ L = $\ell(\hat{p}_\lambda, \mathcal{D}_{cal}) - \ell(\hat{p}_L, \mathcal{D}_{cal}), 0 \quad$ # (n,)
    $\quad$ UCB[i] = WSR(L, $\delta$) $\quad$ # Algorithm 2

```
# Find λ̂_UCB (Eq. 12)
rcp = -1
```
**for** $i,\ ucb$ *in enumerate(UCB[1:])* **do**
    $\quad$ **if** $ucb \; \texttt{>=} \; \epsilon$ **then**
    $\quad\quad$ | rcp = i break

**return** *grid[rcp]*

---

**Algorithm 2:** WSR bound (Prop. 3) for UCB
(see also Section 3.1.3 in Bates et al. [8])

---

**input** : `losses` L, $\delta,$ `grid step` $\Delta,$
$\quad\quad\quad \nu$ `bound` B (default B = 1)
**output:** UCB $\hat{\mathcal{R}}^+(\lambda)$

```
n = len(L)

# init arrays
```
$\mu, \sigma^2, \nu, \kappa$ = `np.ones(n)`, ...
**for** $i$ *in range(n)* **do**
    $\quad \mu[i] = (1/2 + \sum_{j=0}^{i} \text{L}[j])/(i+1)$
    $\quad \sigma^2[i] = (1/4 + \sum_{j=0}^{i}(\text{L}[j] - \mu[j])^2)/(i+1)$
    $\quad \nu[i] = \min\{1/\text{B}, \sqrt{\frac{2\log(1/\delta)}{n\sigma^2[i-1]}}\}$
    $\quad$ # define $\kappa$ function
    $\quad \kappa[i] = \texttt{lambda } \epsilon : \prod_{i=0}^{j}\{1 - \nu[j](\text{L}[j] - \epsilon)\}$
```
grid = np.arange(0,1,Δ)
```
**for** $\epsilon$ *in grid* **do**
    $\quad$ **if** $\max_i \kappa[i](\epsilon) > 1/\delta$ **then**
    $\quad\quad$ | **return** $\epsilon$

---

# C  Implementation Details

Since our approach is *post-hoc*, the required compute resources needed are minimal. The primary requirement is sufficient memory to process the required calibration and test data, which can be quite large (*e.g.*, images of size $2048 \times 1024$ pixels for Cityscapes). All our experiments can be performed and replicated on a single A100 GPU with experiment runtimes of $<1$ day. We detail further case-specific implementations below for each experiment.

## C.1  Image Classification

**Model details.**    The models we consider are: the multi-scale dense network (**MSDNet**; [36]), an adaptation of traditional convolutional NNs for the early-exit setting; the dynamic vision transformer (**DViT**; [74]), which comprises multiple transformers with an increasing number of input patches; an enhanced MSDNet model that weights easy and hard examples differently during training (**L2W-DEN**; [30]); and a recently proposed dynamic perceiver (**Dyn-Perc**; [31]), which decouples feature extraction and early prediction tasks via a novel dual-branch architecture. For all models, we either work with the publicly available pretrained checkpoints or train the models ourselves, closely following the original implementation details.

## C.2  Semantic Segmentation

**Model details.**    We consider the EENN proposed by Liu et al. [48] (ADP-C), which adds three intermediate exit heads to the HRNet segmentation model [73] (for a total of four exits) and is trained end-to-end on Cityscapes [17]. The model comes in two sizes, small (W18) and large (W48). We focus on the larger model (ADP-C-W48), but find results hold equivalently for the smaller one (in fact, larger gains can be obtained). Across experiments we employ the publicly available model checkpoints from the original implementation[5].

**Model finetuning.**    Since the model is trained on Cityscapes, we consider additional finetuning to evaluate ADP-C on GTA5 [59]. We take the available pre-trained model checkpoint (APD-C-W48) and finetune the model for 50 epochs on the GTA5 training set ($\sim 12000$ images). For this purpose, we employ the original training script and training parameters (*e.g.*, learning rate, batch size, *etc.*). However, we find that our finetuned model does not perform on par with the original, *i.e.*, performance on GTA5 is substantially inferior to that on Cityscapes. In particular, the performance improvement across subsequent exits on GTA5 is marginal, resulting in an EENN that is less suitable for early-exiting (see also § D.2). Yet, we find that risk control frameworks still apply, highlighting their robust model-agnostic properties even in light of an inferior underlying predictor.

**Image-level aggregation.**    ADP-C provides an exiting mechanism following Eq. 1 on *pixel-level*, which is less useful for down-stream applications and decision making. For details on the exact mechanism, we refer to Liu et al. [48]. Rather than exiting only for selected image pixels, we instead want to early-exit the entire image whilst ensuring risk control. Thus, alongside different per-pixel confidence scores $\hat{c}_{l,i,j}$, $(i,j) \in H \times W$, we also consider confidence aggregations $\phi(\cdot)$ which produce a single image-level confidence measure $\hat{c}_l$ to perform *image-level* risk control. Note that our prediction losses *mean intersection-over-union* and *miscoverage* already aggregate from pixel- to image-level, whereas our distributional loss (Eq. 7) is adapted to additionally average over pixels, resulting in a *mean-pixel* Brier score interpretation (see § A.3).

**Risk control evaluation.**    We evaluate the original ADP-C-W48 on Cityscapes validation data, with a split of 80% $\mathcal{D}_{cal}$ (*i.e.*, 400 images) and 20% $\mathcal{D}_{test}$ (*i.e.*, 100 images). Similarly, we randomly select a subset of 1000 images from the GTA5 validation set and evaluate our finetuned model using 80% $\mathcal{D}_{cal}$ (*i.e.*, 800 images) and 20% $\mathcal{D}_{test}$ (*i.e.*, 200 images). In both cases, we average risk control results over 100 trials of random calibration and test splits.

---

[5]See https://github.com/liuzhuang13/anytime

## C.3 Language Modeling

**Model details.** For our language modeling experiments, we employ the early-exit pretrained model based on T5-large (770M parameters) from Bae et al. [5][6]. While this model closely follows the implementations in Schuster et al. [65], we found it easier to work with than the original code[7]. Note that Schuster et al. [65] report results for T5-small (60M parameters) and T5-base (220M parameters), whereas we use the larger T5-large. For risk control evaluation, we follow their exact exiting mechanism. Specifically, we compute softmax-based confidences at every exit and deploy their threshold decay mechanism, where early-exiting is more conservative for initial tokens and becomes progressively more permissive ([65], Eq. 5).

## C.4 Image Generation with Early-Exit Diffusion

**Model details.** For our image generation experiment, we re-implement the early-exit diffusion model proposed by Tang et al. [69] (DeeDiff), since the original code is not publicly available. We model our training procedure as closely as possible to the original. As suggested in the paper, we use the U-ViT transformer [7] as a backbone denoising network. Early-exiting in DeeDiff is performed on the denoising network at each sampling step. Specifically, for every sampling step $t$ and exit layer $e = 1, \ldots, L$, a per-pixel confidence map $\boldsymbol{c}_{e,t}$ is obtained. Then, $\boldsymbol{c}_{e,t}$ is used to compute the global (scalar) confidence score $c_{e,t}$ by averaging the confidence across all pixels. If the scalar confidence score satisfies the exit condition $c_{e,t} \geq \lambda$, we proceed to the next denoising step $t + 1$, employing the output (*i.e.*, the predicted noise) of the $e$-th exit layer at the $t$-th sampling step. The model is trained using a standard diffusion denoising loss [35] and two uncertainty-aware losses, closely following the approach described in Tang et al. [69].

**LPIPS metric.** Our task-specific prediction loss measures the *perceptual difference* between early-exit and full model image generations. For this, we employ the LPIPS metric [82], which computes the similarity between activations of image patches for a selected pre-trained network. LPIPS values are in the range of $[0, 1]$, with smaller values indicating perceptually more similar images.

# D  Further Experimental Results

## D.1 Image Classification

Additional test risk and efficiency curves for calibration set size $n = 1000$ on ImageNet are displayed in Fig. 8, while tables with efficiency gain values on ImageNet are given in Table 2.

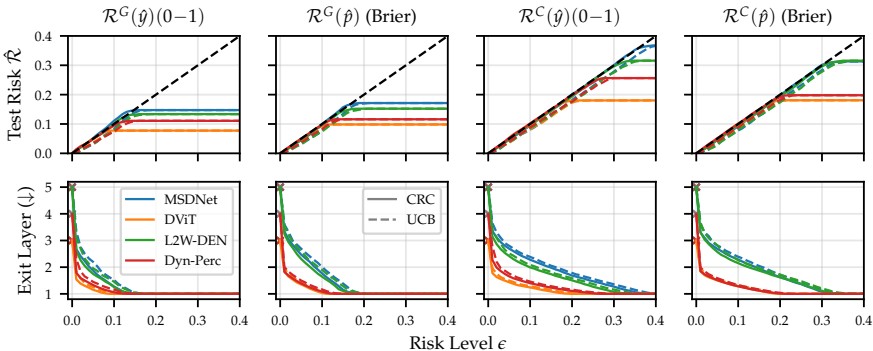

Figure 8: Empirical test risk (*top*) and efficiency gains (*bottom*) for different early-exit models, risks (§ 3.2) and risk levels $\epsilon$ on ImageNet (for calibration set size $n = 1000$). In line with theoretical results, the test risk is controlled across models, risk types, and levels. Despite guaranteeing control *in expectation* (CRC, Prop. 1) or *with high probability* (UCB, Prop. 2), obtained gains are substantial.

---

[6]See https://github.com/raymin0223/fast_robust_early_exit
[7]See https://github.com/google-research/t5x/tree/main/t5x/contrib/calm

Table 2: Efficiency gains for various EENNs on ImageNet, for risk control via CRC (Prop. 1) or UCB (Prop. 2) and calibration set size $n \in \{100, 1000\}$. Displayed values denote relative improvement over last-layer exiting in terms of mean exit layer (in %). The test risk is successfully controlled in all cases. Results focus on small risk levels $\epsilon \in \{0.01, 0.05\}$, which are of higher practical interest.

(a) UCB and $n = 100$

| Risk | $\mathcal{R}^G(\hat{y})(0{-}1)$ | | $\mathcal{R}^G(\hat{p})$ (Brier) | | $\mathcal{R}^C(\hat{y})(0{-}1)$ | | $\mathcal{R}^C(\hat{p})$ (Brier) | |
|---|---|---|---|---|---|---|---|---|
| Level $\epsilon$ | **0.01** | **0.05** | **0.01** | **0.05** | **0.01** | **0.05** | **0.01** | **0.05** |
| MSDNet | 0.0 | 42.51 | 0.0 | 30.62 | 0.0 | 34.21 | 0.0 | 30.47 |
| DViT | 0.0 | 46.5 | 0.0 | 35.4 | 0.0 | 45.14 | 0.0 | 39.74 |
| L2W-DEN | 0.0 | 49.47 | 0.0 | 33.09 | 0.0 | 41.77 | 0.0 | 35.77 |
| Dyn-Perc | 0.0 | 45.42 | 0.0 | 35.67 | 0.0 | 31.71 | 0.0 | 34.85 |

(b) CRC and $n = 100$

| Risk | $\mathcal{R}^G(\hat{y})(0{-}1)$ | | $\mathcal{R}^G(\hat{p})$ (Brier) | | $\mathcal{R}^C(\hat{y})(0{-}1)$ | | $\mathcal{R}^C(\hat{p})$ (Brier) | |
|---|---|---|---|---|---|---|---|---|
| Level $\epsilon$ | **0.01** | **0.05** | **0.01** | **0.05** | **0.01** | **0.05** | **0.01** | **0.05** |
| MSDNet | 41.64 | 58.6 | 10.99 | 43.58 | 30.71 | 43.68 | 4.46 | 40.54 |
| DViT | 49.32 | 58.6 | 5.02 | 51.09 | 40.36 | 51.83 | 4.58 | 46.89 |
| L2W-DEN | 50.82 | 64.38 | 8.09 | 50.4 | 35.09 | 50.57 | 0.0 | 44.66 |
| Dyn-Perc | 44.39 | 63.9 | 36.86 | 59.42 | 30.09 | 57.99 | 31.19 | 58.05 |

(c) UCB and $n = 1000$

| Risk | $\mathcal{R}^G(\hat{\boldsymbol{y}})(0{-}1)$ | | $\mathcal{R}^G(\hat{\boldsymbol{p}})$ (Brier) | | $\mathcal{R}^C(\hat{\boldsymbol{y}})(0{-}1)$ | | $\mathcal{R}^C(\hat{\boldsymbol{p}})$ (Brier) | |
|---|---|---|---|---|---|---|---|---|
| Level $\epsilon$ | **0.01** | **0.05** | **0.01** | **0.05** | **0.01** | **0.05** | **0.01** | **0.05** |
| MSDNet | 33.23 | 56.0 | 27.94 | 46.11 | 27.05 | 44.0 | 28.05 | 43.65 |
| DViT | 38.24 | 58.12 | 34.93 | 52.31 | 34.61 | 50.02 | 34.5 | 48.31 |
| L2W-DEN | 38.11 | 60.87 | 28.96 | 47.97 | 32.04 | 49.71 | 26.46 | 43.6 |
| Dyn-Perc | 42.24 | 63.77 | 49.9 | 62.51 | 30.48 | 56.01 | 50.34 | 60.28 |

(d) CRC and $n = 1000$

| Risk | $\mathcal{R}^G(\hat{y})(0{-}1)$ | | $\mathcal{R}^G(\hat{p})$ (Brier) | | $\mathcal{R}^C(\hat{y})(0{-}1)$ | | $\mathcal{R}^C(\hat{p})$ (Brier) | |
|---|---|---|---|---|---|---|---|---|
| Level $\epsilon$ | **0.01** | **0.05** | **0.01** | **0.05** | **0.01** | **0.05** | **0.01** | **0.05** |
| MSDNet | 46.56 | 62.55 | 33.11 | 50.56 | 33.01 | 46.12 | 34.09 | 46.26 |
| DViT | 48.68 | 61.6 | 39.05 | 54.89 | 40.75 | 52.21 | 38.48 | 49.86 |
| L2W-DEN | 47.73 | 65.04 | 35.09 | 53.81 | 37.08 | 51.7 | 33.76 | 48.48 |
| Dyn-Perc | 55.16 | 66.73 | 54.28 | 65.25 | 43.35 | 58.62 | 53.67 | 62.04 |

### D.2 Semantic Segmentation

We report full tables with efficiency gains across risks and confidence measure combinations for Cityscapes (Table 3) and GTA5 (Table 4) in terms of mean exit layer and GFLOPS. In addition, we display test risk and efficiency curves on both Cityscapes (Fig. 9) and GTA5 (Fig. 10) for risk control via CRC (Prop. 1) and UCB (Prop. 2). The figures are for the simplest combination of **Top-1** pixel-level confidence and **mean** image-level aggregation. Note that the figures for other confidence combinations are similar and thus omitted, with the test risk being controlled in all cases.

**GTA5 results.** We observe that for GTA5 the *performance gap risk* for prediction control $\mathcal{R}^G(\hat{y})$ (mIoU) seems particularly easy to control, with high gains reached even for very strict $\epsilon = 0.01$. This relates to the underlying predictor's generally inferior performance due to finetuning (see § C.2). The obtained model records lower performance and marginal improvements across exit layers, resulting in a small risk that is easy to control. Intuitively, the price paid by exiting early is marginal, since the early-exit layer performs almost on par with the final layer. Thus, the scale of the risk deviates from that of other risks, and more meaningful risk control should consider both improving the underlying predictor, and selecting a different scale of risk levels $\epsilon$.

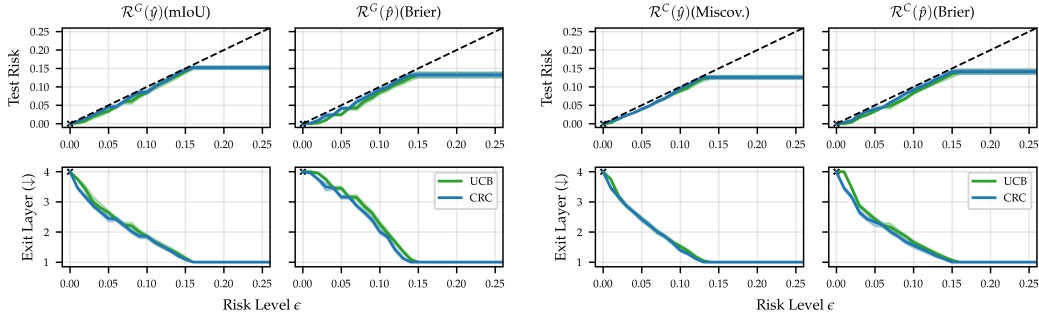

Figure 9: Empirical test risk (*top*) and efficiency gains (*bottom*) for different risks (§ 3.2) and risk levels $\epsilon$ on **Cityscapes**. In line with theoretical results, the test risk is controlled across risk types and levels. Despite guaranteeing control *in expectation* (CRC, Prop. 1) or *with high probability* (UCB, Prop. 2), obtained gains are meaningful. For both figures, we consider the simplest combination of Top-1 confidence score and mean image-level aggregation.

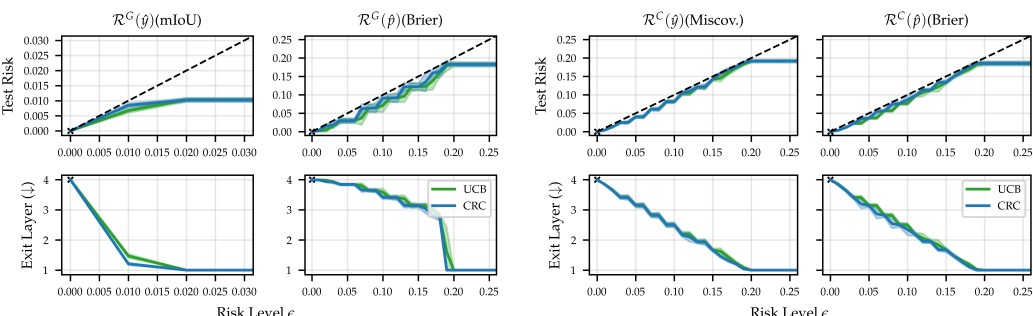

Figure 10: Empirical test risk (*top*) and efficiency gains (*bottom*) for different risks (§ 3.2) and risk levels $\epsilon$ on **GTA5**. In line with theoretical results, the test risk is controlled across risk types and levels. Despite guaranteeing control *in expectation* (CRC, Prop. 1) or *with high probability* (UCB, Prop. 2), obtained gains are meaningful. For both figures, we consider the simplest combination of Top-1 confidence score and mean image-level aggregation.

Table 3: Efficiency gains for semantic segmentation with risk control via UCB (Prop. 2) on **Cityscapes**. We evaluate for different risks (§ 3.2), confidence measures (§ 5.2) and risk levels $\epsilon$. Displayed values denote relative improvement over last-layer exiting (in %) in terms of mean exit layer or floating point operations (GFLOPS). The test risk is successfully controlled in all cases.

(a) Efficiency gains in terms of **mean exit layer** improvement.

| | Risk | $\mathcal{R}^G(\hat{y})$ (mIoU) | | | $\mathcal{R}^G(\hat{p})$ (Brier) | | | $\mathcal{R}^C(\hat{y})$ (Miscov.) | | | $\mathcal{R}^C(\hat{p})$ (Brier) | | |
|---|---|---|---|---|---|---|---|---|---|---|---|---|---|
| | Level $\epsilon$ | 0.01 | 0.05 | 0.1 | 0.01 | 0.05 | 0.1 | 0.01 | 0.05 | 0.1 | 0.01 | 0.05 | 0.1 |
| Mean | Top-1 | 6.3 | 33.7 | 53.5 | 0.0 | 13.6 | 43.4 | 6.3 | 39.2 | 61.8 | 0.0 | 39.3 | 58.4 |
| Mean | Top-Diff | 9.3 | 35.5 | 54.4 | 0.0 | 17.5 | 44.3 | 6.3 | 39.9 | 62.4 | 0.0 | 38.6 | 57.9 |
| Mean | Entropy | 5.2 | 36.0 | 54.3 | 0.0 | 17.9 | 41.0 | 5.1 | 40.4 | 61.3 | 0.0 | 40.1 | 58.3 |
| Quant. | Top-1 | 0.0 | 36.7 | 54.6 | 0.0 | 14.9 | 45.0 | 0.0 | 41.2 | 63.4 | 0.0 | 39.1 | 59.4 |
| Quant. | Top-Diff | 0.1 | 37.1 | 55.2 | 0.0 | 17.2 | 45.2 | 0.0 | 41.2 | 63.7 | 0.0 | 40.4 | 59.6 |
| Quant. | Entropy | 6.1 | 37.0 | 54.0 | 0.0 | 17.9 | 44.7 | 6.1 | 41.0 | 63.1 | 0.0 | 39.1 | 58.7 |
| Patch | Top-1 | 10.0 | 35.7 | 53.3 | 0.0 | 18.4 | 45.3 | 8.8 | 39.1 | 61.5 | 0.0 | 38.0 | 58.3 |
| Patch | Top-Diff | 10.0 | 35.2 | 53.4 | 0.0 | 19.4 | 45.9 | 8.8 | 40.5 | 62.2 | 0.0 | 38.4 | 58.8 |
| Patch | Entropy | 9.1 | 34.8 | 53.5 | 0.0 | 18.0 | 45.8 | 8.1 | 38.9 | 61.5 | 0.0 | 37.3 | 57.1 |

(b) Efficiency gains in terms of **GFLOPS** improvement.

| | Risk | $\mathcal{R}^G(\hat{y})$ (mIoU) | | | $\mathcal{R}^G(\hat{p})$ (Brier) | | | $\mathcal{R}^C(\hat{y})$ (Miscov.) | | | $\mathcal{R}^C(\hat{p})$ (Brier) | | |
|---|---|---|---|---|---|---|---|---|---|---|---|---|---|
| | Level $\epsilon$ | 0.01 | 0.05 | 0.1 | 0.01 | 0.05 | 0.1 | 0.01 | 0.05 | 0.1 | 0.01 | 0.05 | 0.1 |
| Mean | Top-1 | 6.2 | 33.4 | 53.2 | 0.0 | 13.5 | 43.1 | 6.2 | 38.9 | 61.4 | 0.0 | 39.0 | 58.0 |
| Mean | Top-Diff | 9.2 | 35.3 | 54.0 | 0.0 | 17.4 | 44.0 | 6.3 | 39.6 | 62.0 | 0.0 | 38.3 | 57.5 |
| Mean | Entropy | 5.1 | 35.7 | 53.9 | 0.0 | 17.7 | 40.7 | 5.1 | 40.1 | 60.9 | 0.0 | 39.8 | 57.9 |
| Quant. | Top-1 | 0.0 | 36.4 | 54.2 | 0.0 | 14.8 | 44.6 | 0.0 | 40.9 | 62.9 | 0.0 | 38.8 | 58.9 |
| Quant. | Top-Diff | 0.1 | 36.8 | 54.8 | 0.0 | 17.1 | 44.8 | 0.0 | 40.9 | 63.3 | 0.0 | 40.1 | 59.1 |
| Quant. | Entropy | 6.0 | 36.7 | 53.6 | 0.0 | 17.8 | 44.4 | 6.0 | 40.7 | 62.7 | 0.0 | 38.8 | 58.2 |
| Patch | Top-1 | 9.9 | 35.4 | 52.9 | 0.0 | 18.2 | 44.9 | 8.7 | 38.8 | 61.0 | 0.0 | 37.7 | 57.8 |
| Patch | Top-Diff | 9.9 | 34.9 | 53.0 | 0.0 | 19.2 | 45.6 | 8.7 | 40.2 | 61.7 | 0.0 | 38.1 | 58.4 |
| Patch | Entropy | 9.1 | 34.6 | 53.1 | 0.0 | 17.8 | 45.5 | 8.0 | 38.6 | 61.1 | 0.0 | 37.0 | 56.7 |

Table 4: Efficiency gains for semantic segmentation with risk control via UCB (Prop. 2) on **GTA5**. We evaluate for different risks (§ 3.2), confidence measures (§ 5.2) and risk levels $\epsilon$. Displayed values denote relative improvement over last-layer exiting (in %) in terms of mean exit layer or floating point operations (GFLOPS). The test risk is successfully controlled in all cases.

(a) Efficiency gains in terms of **mean exit layer** improvement.

| | Risk | $\mathcal{R}^G(\hat{y})$ (mIoU) | | | $\mathcal{R}^G(\hat{p})$ (Brier) | | | $\mathcal{R}^C(\hat{y})$ (Miscov.) | | | $\mathcal{R}^C(\hat{p})$ (Brier) | | |
|---|---|---|---|---|---|---|---|---|---|---|---|---|---|
| | Level $\epsilon$ | 0.01 | 0.05 | 0.1 | 0.01 | 0.05 | 0.1 | 0.01 | 0.05 | 0.1 | 0.01 | 0.05 | 0.1 |
| Mean | Top-1 | 63.3 | 75.0 | 75.0 | 0.2 | 4.0 | 10.3 | 3.9 | 21.2 | 37.4 | 3.7 | 21.5 | 37.6 |
| Mean | Top-Diff | 62.9 | 75.0 | 75.0 | 0.3 | 4.6 | 12.0 | 4.4 | 21.8 | 39.2 | 3.0 | 23.6 | 43.0 |
| Mean | Entropy | 62.5 | 75.0 | 75.0 | 0.2 | 2.9 | 12.5 | 2.8 | 18.5 | 39.9 | 2.8 | 18.7 | 42.9 |
| Quant. | Top-1 | 63.4 | 75.0 | 75.0 | 0.0 | 4.6 | 12.4 | 2.5 | 23.1 | 42.3 | 2.4 | 23.4 | 42.6 |
| Quant. | Top-Diff | 63.8 | 75.0 | 75.0 | 0.0 | 4.5 | 12.6 | 4.2 | 22.8 | 42.1 | 3.0 | 24.1 | 43.4 |
| Quant. | Entropy | 61.1 | 75.0 | 75.0 | 0.1 | 4.9 | 14.0 | 3.7 | 23.6 | 41.8 | 3.6 | 23.9 | 43.6 |
| Patch | Top-1 | 60.1 | 75.0 | 75.0 | 2.9 | 18.3 | 35.6 | 3.9 | 18.9 | 35.5 | 2.9 | 18.3 | 35.6 |
| Patch | Top-Diff | 60.2 | 75.0 | 75.0 | 3.5 | 19.7 | 37.2 | 4.7 | 19.5 | 37.1 | 3.5 | 19.7 | 37.2 |
| Patch | Entropy | 58.9 | 75.0 | 75.0 | 2.2 | 19.0 | 36.7 | 3.8 | 19.0 | 36.5 | 2.2 | 19.0 | 36.7 |

(b) Efficiency gains in terms of **GFLOPS** improvement.

| | Risk | $\mathcal{R}^G(\hat{y})$ (mIoU) | | | $\mathcal{R}^G(\hat{p})$ (Brier) | | | $\mathcal{R}^C(\hat{y})$ (Miscov.) | | | $\mathcal{R}^C(\hat{p})$ (Brier) | | |
|---|---|---|---|---|---|---|---|---|---|---|---|---|---|
| | Level $\epsilon$ | 0.01 | 0.05 | 0.1 | 0.01 | 0.05 | 0.1 | 0.01 | 0.05 | 0.1 | 0.01 | 0.05 | 0.1 |
| Mean | Top-1 | 62.8 | 75.0 | 75.0 | 0.2 | 3.9 | 10.2 | 3.9 | 21.1 | 37.2 | 3.7 | 21.3 | 37.3 |
| Mean | Top-Diff | 62.5 | 75.0 | 75.0 | 0.3 | 4.6 | 11.9 | 4.4 | 21.6 | 38.9 | 3.0 | 23.5 | 42.7 |
| Mean | Entropy | 62.0 | 75.0 | 75.0 | 0.2 | 2.9 | 12.4 | 2.8 | 18.3 | 39.6 | 2.8 | 18.6 | 42.6 |
| Quant. | Top-1 | 63.0 | 75.0 | 75.0 | 0.0 | 4.5 | 12.4 | 2.5 | 22.9 | 42.0 | 2.4 | 23.2 | 42.3 |
| Quant. | Top-Diff | 63.4 | 75.0 | 75.0 | 0.0 | 4.5 | 12.5 | 4.2 | 22.7 | 41.7 | 3.0 | 24.0 | 43.1 |
| Quant. | Entropy | 60.6 | 75.0 | 75.0 | 0.1 | 4.9 | 13.9 | 3.7 | 23.4 | 41.5 | 3.5 | 23.7 | 43.2 |
| Patch | Top-1 | 59.7 | 75.0 | 75.0 | 2.9 | 18.2 | 35.4 | 3.9 | 18.7 | 35.2 | 2.9 | 18.2 | 35.4 |
| Patch | Top-Diff | 59.8 | 75.0 | 75.0 | 3.4 | 19.5 | 36.9 | 4.7 | 19.4 | 36.8 | 3.4 | 19.5 | 36.9 |
| Patch | Entropy | 58.4 | 75.0 | 75.0 | 2.2 | 18.8 | 36.4 | 3.7 | 18.8 | 36.2 | 2.2 | 18.8 | 36.4 |

## D.3 Language Modeling

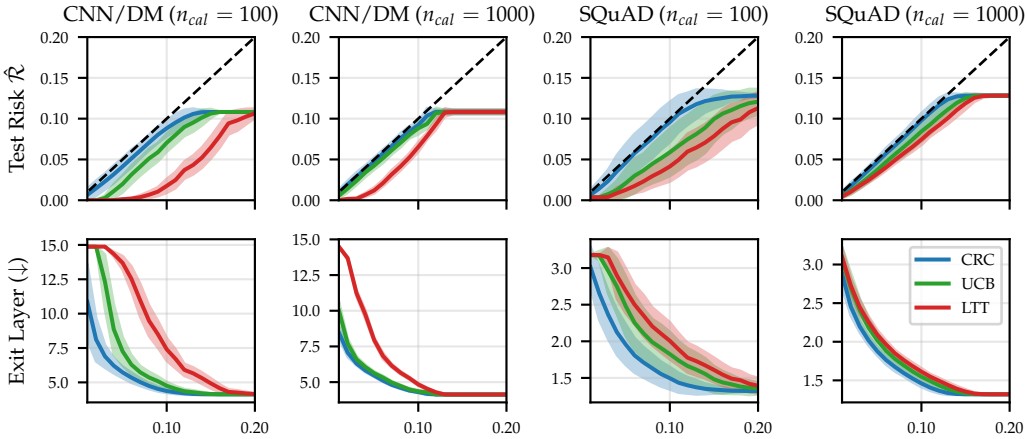

Figure 11: Empirical test risk (*top*) and efficiency gains (*bottom*) for the CALM model [65] for text summarization on CNN/DM and question answering on SQuAD. Our adaptation of UCB [8] (Prop. 2) outperforms the LTT [1] approach in CALM by yielding larger efficiency gains under the same risk control assurances. Shading denotes the standard deviation across $S = 100$ calibration/test splits.

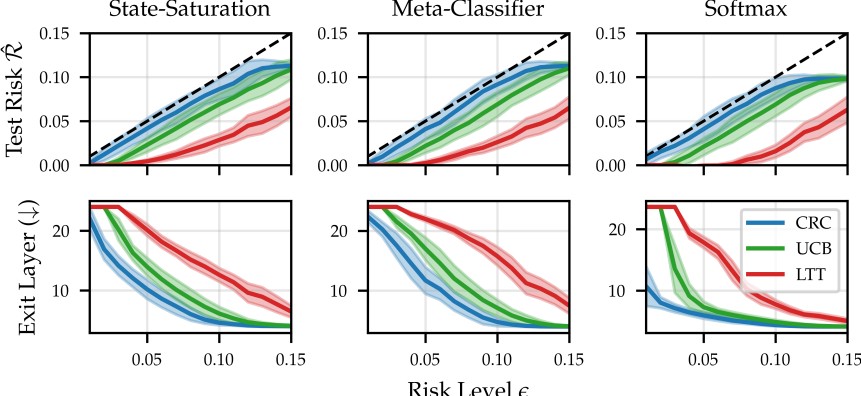

Figure 12: Empirical test risk (*top*) and efficiency gains (*bottom*) for the CALM model for text summarization on CNN/DM across different confidence measures (see Schuster et al. [65], §3.5). *From left to right*: Hidden state saturation, meta-classifiers, and top-2 softmax difference. Our employed risk control frameworks based on CRC and UCB continue to outperform LTT across all measures of confidence. Shading denotes the standard deviation across $S = 100$ splits.

## D.4 Image Generation with Early-Exit Diffusion

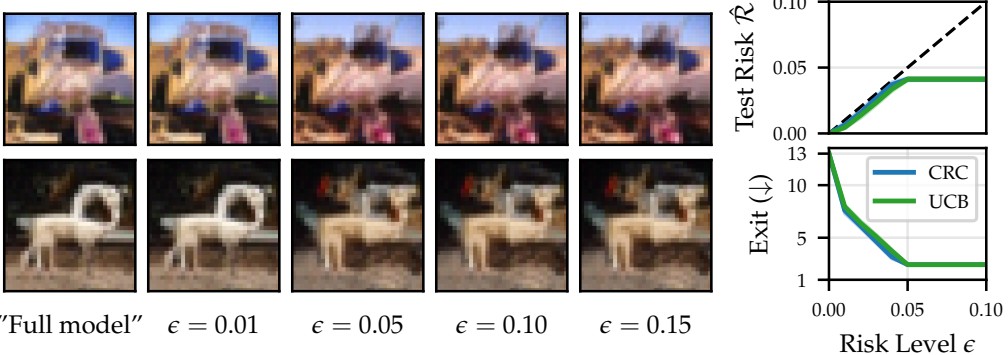

Figure 13: Results for early-exit diffusion with DeeDiff [69] on CIFAR [42]. *Right:* Empirical test risks are controlled for both CRC (Prop. 1) and UCB (Prop. 2) (for calibration set size $n = 500$). *Left:* The quality of generated images is directly related to the targeted risk control level $\epsilon$.

