# OpenReview forum: "Fast yet Safe: Early-Exiting with Risk Control"
_NeurIPS.cc/2024/Conference — NeurIPS 2024 poster_

### Official Review · Reviewer_q2CX · 2024-06-26

**Soundness:** 3
**Presentation:** 3
**Contribution:** 3
**Rating:** 6
**Confidence:** 3

**Summary:**

This paper investigate how to adapt frameworks of risk control to early-exit neural networks (EENN). The new framework is designed to exit without severely degrading performance as well as to guarantee the reliability of the prediction during early exiting.  The applied risk control technique offers a distribution-free, post-hoc solution that tunes the EENN’s exiting mechanism so that exits only occur when the output is of sufficient quality. Experiments on a series of tasks, including image classification, semantic segmentation, language modelling, and image generation show the effectiveness of the proposed method.

-------------after rebuttal----------

The authors address most of my concerns. Although the single exiting threshold seems somehow a little flaw, the proposed new method is interesting and new.

**Strengths:**

1. The experiments on different vision tasks can be strong evidence to show the effectiveness of the proposed method.
2. The theoretical analysis seems reasonable to be.

**Weaknesses:**

The main weakness of this paper is under the strong assumption that all exiting thresholds are set to the same values. I admit that relaxing this assumption by adopting RC techniques for a high-dimensional threshold can lead to some difficulties in theoretical analysis, while using the same threshold for all exits may harm the overall performance of EENN. Only empirical test risk is used in all experiments. The general performance of EENN models in different tasks is also an important evaluation metric for EENN.

The applicability of the proposed RC method can be impeded if the prediction performance is unsatisfactory when using the same exiting threshold for every exit classifiers, even though it has a lower risk in the prediction.

It is also suggested to review the related important works in early exiting neural network areas, such as:
[1] Improved techniques for training adaptive deep networks. in ICCV, 2019.
[2] Resolution Adaptive Networks for Efficient Inference, in CVPR, 2020.
[3] Learning to Weight Samples for Dynamic Early-exiting Networks, in ECCV, 2022.

**Questions:**

See weakness. I think more evaluation metrics, such as test accuracy for classification tasks, or the IOU for segmentation tasks, should be included to show the effectiveness of the proposed method.

**Limitations:**

See weakness. I think it is acceptable to use the same threshold for theoretical analysis. However, the real performance of the evaluated EENNs should also be attached in the manuscript.

---

> ### Author Rebuttal · Authors · 2024-08-06
>
> Dear reviewer q2CX, we thank you for your time and helpful comments, and address your two key concerns in the following.
>
> > The main weakness of this paper is under the strong assumption that all exiting thresholds are set to the same values. I admit that relaxing this assumption by adopting RC techniques for a high-dimensional threshold can lead to some difficulties in theoretical analysis, while using the same threshold for all exits may harm the overall performance of EENN.
>
> > The applicability of the proposed RC method can be impeded if the prediction performance is unsatisfactory when using the same exiting threshold for every exit classifiers, even though it has a lower risk in the prediction.
>
> Please consider our response under point (1) in the general rebuttal to this question, and kindly provide additional questions in case you feel it has not been sufficiently addressed.
>
> > Only empirical test risk is used in all experiments. The general performance of EENN models in different tasks is also an important evaluation metric for EENN.
>
> > I think more evaluation metrics, such as test accuracy for classification tasks, or the IOU for segmentation tasks, should be included to show the effectiveness of the proposed method.
>
> > However, the real performance of the evaluated EENNs should also be attached in the manuscript.
>
> Thanks for raising these points as they made us realise there is some misunderstanding about the interpretation of our proposed risks, and their relationship to standard performance metrics. As we explicitly state in L164-168 and seen from the equations in §3.2, our risks are formulated as the mean relative performance difference between full (i.e., last-layer exit) and early-exit model, and provide a generic template in which we may plug in different labels, model outputs, confidence measures, and loss functions.
>
> We explore a wide range of loss functions across tasks for prediction control, and in particular **use standard task-specific performance losses** (and precisely the ones you mention): 0-1 loss for classification, mIOU loss and miscoverage loss for segmentation, ROUGE-L loss and Token-F1 loss for language modelling, and LPIPS loss for image generation. Thus, the empirical test risk equates the difference in test accuracy between the full and early-exit model for our classification experiments in §5.1, the difference in mIOU between the full and early-exit model for some of our segmentation experiments in §5.2, etc., providing interpretable assessments of the EENN’s performance based on risk control.
>
> We would also emphasis here that our risk control approach is entirely *post-hoc*. As such, **we do not change the performance of any underlying model but work with existing pretrained EENNs**, and thus our model performances are equivalent to the numbers provided by the original authors. Thus, we consider reporting the commonly seen performance vs. FLOP curves found in early-exit papers as redundant for our work, since we do not modify the underlying models (except for switching to a single exit threshold for some models like MSDNet, see our point (1) in meta rebuttal for a longer discussion on this).
>
> Perhaps it's also helpful to add here that we may interpret our risk control threshold selection as a framework to navigate the performance vs. FLOP curve based on the user’s performance requirements, i.e. we find the point on the curve such that the EENN’s test performance is guaranteed to be at most $\epsilon$ (e.g., at most 5%) worse than the full model (i.e., last-layer exit). As such, evaluating against different risk levels $\epsilon$ is the appropriate way to benchmark our procedure, and provided “test risk” figures such as Fig. 2 and 3 in the paper are common in the risk control domain (see, e.g., Fig. B.1 in [2]). Moreover, the fact that the test risk curves are close to the diagonal (e.g., see Figure 3, upper row) is encouraging, as it suggests that with our framework, there is very little efficiency cost for the added safety of having risk-control guarantees.
>
> We thank you for raising these points, highlighting the need to better explain the provided test risk figures. We attempted to do so in §5 (L272-278), but shall expand upon it for the camera-ready version. Likewise, we will aim to clarify that model performance is not affected by our post-hoc risk control procedure.
>
> [2] Schuster, T., et al. Confident adaptive language modeling. *NeurIPS 2022*
>
> > It is also suggested to review the related important works in early exiting neural network areas, such as: [1] Improved techniques for training adaptive deep networks. in ICCV, 2019. [2] Resolution Adaptive Networks for Efficient Inference, in CVPR, 2020. [3] Learning to Weight Samples for Dynamic Early-exiting Networks, in ECCV, 2022.
>
> We appreciate the additional references. We would politely point out that reference [3] is already contained in our image classification experiments, where we refer to it as L2W-DEN (see Figure 3 in the paper). Following your advice, **we have also added the suggested reference [1] to our experiments**. We refer to the model as IMTA and observe similar results to the other models, see Figure A of the attached rebuttal PDF. We will incorporate this model alongside your other suggested reference [2] (RANet) for the camera-ready version (the ImageNet training of the RANet unfortunately didn't finish before the rebuttal deadline), either into the main figure or into an appendix figure to preserve readability.
>
> We thank you again for your comments and would appreciate you considering raising your score if you believe they have been addressed sufficiently.

---

> > ### Author Response · Authors · 2024-08-11
> >
> > Thanks for updating your review, we are glad to hear we were able to address most of your concerns. We also appreciate you increasing your score and will make sure to include a longer discussion on the single-threshold assumption in the camera-ready version. In case any additional questions arise during the discussion period, we would be happy to address them.

---

### Official Review · Reviewer_LhAs · 2024-06-26

**Soundness:** 3
**Presentation:** 3
**Contribution:** 3
**Rating:** 6
**Confidence:** 4

**Summary:**

This paper introduces a risk control framework for early-exit neural networks to balance the trade-off between inference efficiency and model performance. Compared to the conventional methods that manually set the confidence thresholds for early exiting, this work proposes a method to determine exit points from the perspective of risk control. Experiments on vision and language tasks demonstrate the effectiveness.

**Strengths:**

* Novelty: The integration of risk control into EENNs is novel, and the studied problem is an interesting and important challenge in deploying early-exiting models;
* Empirical Validation: The paper provides extensive experimental evidence across multiple domains, demonstrating the practical applicability of the proposed methods;
* Soundness: The risk control framework is well-developed, offering both theoretical foundations and practical algorithms for determining exit thresholds.

**Weaknesses:**

Overall I like the paper. A small suggestion is to demonstrate the accuracy-FLOPs curves as in the compared methods (MSDNet, L2W-DEN, and Dyn-Perceiver), because the current x-axis, "risk level", might be insufficiently straightforward.

**Questions:**

Please see weaknesses.

**Limitations:**

yes

---

> ### Author Rebuttal · Authors · 2024-08-06
>
> Dear reviewer LhAs, we thank you for your time and comments, and address your raised point in the following.
>
> > A small suggestion is to demonstrate the accuracy-FLOPs curves as in the compared methods (MSDNet, L2W-DEN, and Dyn-Perceiver), because the current x-axis, "risk level", might be insufficiently straightforward
>
> We would politely point out here that our risk control approach is entirely *post-hoc*. As such, we do not change the performance of any underlying model but work with existing pretrained EENNs, and thus our model performances are equivalent to the numbers provided by the original authors. While we provide performance curves in Fig. 1 for some models, this was to stress the marginal monotonicity condition rather than testing model’s ability. Thus, we consider reporting the commonly seen performance vs. FLOP curves found in early-exit papers as redundant for our work, since we do not modify the underlying models.
>
> Perhaps it's also helpful to add here that we may interpret our risk control threshold selection as a framework to navigate the performance vs. FLOP curve based on the user’s performance requirements, i.e. we find the point on the curve such that the EENN’s test performance is *guaranteed* to be at most $\epsilon$ (e.g., at most 5%) worse than the full model (i.e., last-layer exit). Hence, we believe that evaluating against different risk levels $\epsilon$ is the appropriate way to benchmark our procedure. Also note that provided “test risk” figures such as Fig. 2 and 3 in the paper are standard in the risk control domain (see, e.g., Fig. B.1 in [2]). Moreover, the fact that the test risk curves are close to the diagonal (e.g., see Figure 3, upper row) is encouraging, as it suggests that with our framework, there is very little efficiency cost for the added safety of having risk-control guarantees.
>
> We thank you for the suggestion, highlighting the need to better explain provided test risk figures. We attempted to do so in §5 (L272-278), but shall expand upon it for the camera-ready version based on your feedback.
>
> [2] Schuster, T., et al. Confident adaptive language modeling. *NeurIPS 2022*

---

> ### Comment · Reviewer_LhAs · 2024-08-08
>
> Thanks for replying. I shall maintain my rating.

---

> > ### Author Response · Authors · 2024-08-09
> >
> > Dear reviewer LhAs, we appreciate your acknowledgement and consideration. In case any additional questions arise during the discussion period or can be clarified in order to improve your rating, we would be happy to address those.

---

### Official Review · Reviewer_c7rs · 2024-07-11

**Soundness:** 2
**Presentation:** 2
**Contribution:** 2
**Rating:** 4
**Confidence:** 3

**Summary:**

This paper proposes a method for improving the efficiency of early-exit neural networks (EENNs) while maintaining performance. ​ EENNs allow for predictions to be made at intermediate layers, resulting in faster inference. ​ However, the challenge is determining when it is safe to exit without sacrificing performance. ​ The authors address this issue by adapting risk control frameworks to EENNs. ​ They propose risk functions to measure the performance drop resulting from early-exiting and demonstrate the effectiveness of their approach on various vision and language tasks. ​

**Strengths:**

- The paper addresses an important problem in machine learning - improving the efficiency of inference without sacrificing performance. ​
- The authors propose a novel approach of adapting risk control frameworks to EENNs, which provides a post-hoc solution for determining when it is safe to exit. ​
- The paper provides empirical validation of their approach on a range of vision and language tasks, demonstrating substantial computational savings while preserving performance goals. ​
- The authors consider both prediction quality and uncertainty estimation in their risk control framework, which is important for safety-critical scenarios. ​

**Weaknesses:**

- The empirical validation of the approach is limited to a range of vision and language tasks. It would be beneficial to see the application of the method to other domains as well.
- The paper does not discuss the limitations or potential drawbacks of the proposed approach. It would be helpful to have a discussion on the potential trade-offs or challenges in implementing the risk control framework in practice.

**Questions:**

Refer to Weaknesses.

---

> ### Author Rebuttal · Authors · 2024-08-06
>
> Dear reviewer c7rs, we thank you for your time and comments, and address your two raised concerns in the following.
>
> > The empirical validation of the approach is limited to a range of vision and language tasks. It would be beneficial to see the application of the method to other domains as well.
>
> We politely disagree with this statement. Our experiments incorporate a wide variety of models, confidence measures and risk types across multiple machine learning tasks, datasets and data modalities (image classification on ImageNet with MSDNet, DViT, L2W-DEN and Dyn-Perc for 0—1 and Brier losses; semantic segmentation on Cityscapes & GTA5 with ADP-C for mIoU, miscoverage and Brier losses and several confidence measures; text summarization on CNN/DM and question answering on SQuAD with T5 for ROUGE-L and Token-F1 losses; image generation with early-exit diffusion on CelebA and CIFAR for LPIPS loss).  Moreover, our extensive experiments have been highlighted as a key strength by all other reviewers (qAvq: "*experimentally demonstrated on a wide range of predictive ... and generative tasks*"; LhAs: "*extensive experimental evidence across multiple domains*"; q2CX: "*The experiments on different vision tasks can be strong evidence to show the effectiveness…*”).
>
> Lastly, we are surprised since your review also lists our experimental evaluation as a strength (”*The paper provides empirical validation of their approach on a range of vision and language tasks, demonstrating substantial computational savings while preserving performance goals*”); perhaps one of the mentions of our experiments was in error?
>
> If you have any concrete additional experiments in mind, please let us know and we would be happy to consider them.
>
> > The paper does not discuss the limitations or potential drawbacks of the proposed approach. It would be helpful to have a discussion on the potential trade-offs or challenges in implementing the risk control framework in practice.
>
> We politely point to §6, which is titled “Conclusion, Limitations and Future Work”.  There we state our main limitation (a single exit threshold) and point to several limitations and opportunities for future work that are rooted in risk control itself, rather than our adaptation to the early-exit setting.  Furthermore, we are explicit about our marginal monotonicity requirement for the EENN (e.g., in our propositions) and discuss it throughout the paper. The presence of Limitations has also been acknowledged by the other reviewers (qAvq: "*Limitations and future work are adequately discussed in the manuscript.*"). Lastly, we will use the additional page of the camera-ready version to further expand on the limitations.
>
> > 4: Borderline reject: Technically solid paper where reasons to reject, e.g., limited evaluation, outweigh reasons to accept, e.g., good evaluation. Please use sparingly.
>
> Based on your current review, it is not entirely clear to us why you are recommending rejection. We would very much appreciate further elaboration of your concerns, or a consideration to raise your score if you believe they have been addressed sufficiently.

---

> > ### Comment · Reviewer_c7rs · 2024-08-13
> >
> > Thanks for the response. I will keep my rating.

---

> ### Author Response · Authors · 2024-08-12
>
> Dear reviewer c7rs, given that your recommendation is in conflict with the other reviewers' and we believe to have addressed the weaknesses raised in your original review, an acknowledgment or response to our rebuttal would be much appreciated. We are very much open to address any concerns in the remaining discussion period.

---

> ### Author Response · Authors · 2024-08-13
>
> Thanks for your acknowledgment of our rebuttal. We would be interested to know which concern that you brought up made you decide to keep your current score? Based on your review and response, this is not clear to us at the moment, and getting a clarity on this would help us a lot in improving our work. We are particularly interested in which part of our experimental setting do you find unsatisfactory and would appreciate any concrete suggestions for improving it.
>
> Thanks in advance!

---

### Official Review · Reviewer_qAvq · 2024-07-12

**Soundness:** 3
**Presentation:** 3
**Contribution:** 3
**Rating:** 7
**Confidence:** 3

**Summary:**

This manuscript revisits the confidence-based thershold tuning for early-exit networks following a risk-control formulation. The proposed approach aims to provide an enhanced mechanism to identify confident predictions early-on during the computation, and exit early to improve inference efficiency with minimal effect on accuracy. The underlying trade-off between accuarcy and latency is balanced by a post-hoc solution, that determines the thershold value for the exits, through a risk control formulation considering a labelled calibration set (Performance Gap Risk), or the predictions of the final exit of the original model (Consistency Risk).

**Strengths:**

- The manuscript studies a very interesting and timely problem, often overlook by the booming filed of early-exit models.
- The proposed approach is post-hoc and can be applied to existing approaches without the need of re-training/finetunning the model, and does not add any computational overhead at inference time.
- The effectiveness of the proposed approach is experimentally demonstrated on a wide range of predictive (image classification and segmentation) and generative tasks (language modelling and image generation), proving the generality of the proposed approach.
- The paper is well-written and offers adequate background information and a nice formulation.

**Weaknesses:**

- The reliance of the proposed approach to marginal monotonicity between performance and depth can be a limiting factor in some applications. E.g. recent works on EE-based LLMs indicate the predictive accuracy fluctuates notably across exits (see Elhoushi et al, LayerSkip, 2024 - Fig.2).
- As stated in the limitations, the adoption of a single and static threshold across exits also significantly reduces the search space of the proposed approach. The inefficiency imposed by this design choice has not been quantified, but is likely to be significant without any calibratuon technics on exit confidence or special training of the model.
- It is unclear how the proposed approach can be applied to the emerging line of works with learnable exit policies, as well as whether it will maintain its performance on different confidence metrics, such as state-saturation, used in LLMs etc

**Questions:**

- Please consider the comments raised on the Weaknesses section and share any insights you may have on them.
- Can the proposed Consistency Risk formulation be used for online adaptation of the thresholds, as during runtime calibration labels are not available, but input distribution shifts may occur?

**Post-Rebuttal Edit:**
Provisionally increasing my score from WA to A, given the thorough clarifications provided by the authors that adequately addressed most of my concerns.

**Limitations:**

Limitations and future work are adequately discussed in the manuscript.

---

> ### Author Rebuttal · Authors · 2024-08-06
>
> Dear reviewer qAvq, we thank you for your time and engaged questions.
>
> > The reliance of the proposed approach to marginal monotonicity between performance and depth can be a limiting [...] (see Elhoushi et al, LayerSkip, 2024 - Fig.2).
>
> This is a great point and we fully agree that for a particular sample the predictive accuracy can fluctuate across exits. We refer to this effect as *overthinking* (following [5, 12]) and mention it several times throughout the paper (L77-80, L196-198, L211-214). However, note that our approaches do not require such strong assumptions on *conditional* monotonicity, i.e., monotonic behaviour per sample. Rather, we only require the milder assumption of *marginal* monotonicity, i.e., that predictive accuracy is monotone across exits *on average* over samples (Eq. 2 in the paper). Such a marginal condition is often implicitly assumed in the early-exit literature, and validated by the commonly shown performance vs. FLOP curves which monotonically grow as the budget increases. Indeed, Fig. 2 in Elhoushi et al. [13] is showcasing a lack of *conditional* monotonicity, since the model outputs are shown for a particular input prompt. Looking at Fig. 6 and 8 from the same paper, we observe that performance is generally monotonic w.r.t. exits on average (marginally). In fact, **the difference between conditional and marginal monotonicity serves as a motivation for one of our main technical contributions** - in the original formulation, CRC requires conditional monotone risks, while we extend this to marginal monotone ones (see Prop. 1). As we state in the paper (L192-198), such an extension is crucial to make CRC amenable to EENNs because of the overthinking issue.
>
> > As stated in the limitations, the adoption of a single and static threshold across exits also significantly reduces the search space [...] inefficiency imposed by this design choice has not been quantified... .
>
> Please consider our response under point (1) in the general rebuttal, and kindly provide additional questions in case you feel it has not been sufficiently addressed.
>
> > how the proposed approach can be applied to the emerging line of works with learnable exit policies
>
> We assume you refer to papers such as JEI-DNN [14], where exits are determined via direct modeling of exit probabilities. If so, it is correct that our procedure does not directly transfer since we require a threshold-based exiting mechanism. We will ensure to better define the scope of our contribution, and we thank you for pointing this out. Note, however, that models with learnable exit policies are less adaptive compared to their threshold counterparts, as they require retraining the model from scratch for every new computational budget. In contrast, threshold-based models can be adapted in a post-hoc fashion by simply tuning the threshold.
>
> Let us also add here that thresholding is not used only in early-exiting, but also in some other emerging lines of work like (soft) speculative decoding [15], where the rollback policy is governed by thresholding. Our risk control framework could be applied to such settings as well, and we will try to add an experiment to the camera-ready version to better exemplify the scope of our contribution.
>
> > whether it will maintain its performance on different confidence metrics, such as state-saturation, used in LLMs
>
> We note that **our approach is agnostic to the choice of confidence measure** used when early-exiting. We aim to demonstrate this in the paper with our segmentation experiment in §5.2 (see Table 1; and Table 3 in §D.2), where we consider different confidence measures and aggregators. Based on your comment, **we have extended our language modeling experiment to include additional confidence measures**: (i) hidden-state saturation and (ii) meta-classifiers, as proposed by [2]. We observe in the Figure B of the attached rebuttal PDF that our employed risk control frameworks based on CRC and UCB continue to outperform LTT across all confidence measures, and will include these results in the camera-ready version.
>
> > Can the proposed Consistency Risk formulation be used for online adaptation [...], but input distribution shifts may occur?
>
> We are not entirely sure what form of online adaptation is meant here, so kindly inform us if the question was not properly addressed.
>
> For distribution shifts between training and calibration data (i.e., $P_{train} \neq P_{cal}$) risk control continues to be applicable as long as the model’s marginal monotonicity is not violated. Our post-hoc approach treats the EENN as a black-box and makes no assumptions on the training distribution (as stated in §2, L50-51). For shifts between calibration and test data (i.e., $P_{cal} \neq P_{test}$) the provided guarantees cease to hold, since an *i.i.d.* assumption on those samples is made. However, under benign shifts the empirical test risks may continue to be controlled.
>
> Since we guarantee risk control even for small calibration set sizes ($n \approx 100$) and are computationally light-weight, a naive online update could see a periodical re-calibration of the exit threshold based on collected test samples, which, in the case of our *consistency risk*, would indeed not even require any test labels. We fully agree that it is an interesting direction for future work to explore proper online updating.
>
> We thank you for your comments and would appreciate you considering raising your score if you believe they have been addressed sufficiently.
>
> **References**
>
> [12] Jazbec, M., et al. Towards anytime classification in early-exit architectures by enforcing conditional monotonicity. *NeurIPS 2023*
>
> [13] Elhoushi, Mostafa, et al. Layer skip: Enabling early exit inference and self-speculative decoding. *arXiv preprint* (2024)
>
> [14] Regol, F., et al. Jointly-learned exit and inference for a dynamic neural network: Jei-dnn. *ICLR 2024*
>
> [15] Kim, S., et al. Speculative decoding with big little decoder. *NeurIPS 2023*

---

> > ### Comment · Reviewer_qAvq · 2024-08-09
> >
> > Thank you very much for the thorough and insightful reply. Most of my raised concerns have been adequately addressed, and as such, I am inclined to provisionally increase my score to Accept, pending the upcoming reviewer discussion.
> >
> > I would suggest that the discussion about the applicability of the proposed approach to multi-exit models with different thresholds in each exit (where the search space is notably harder to explore) is extended in the manuscript, along the lines of the relevant discussion in the rebuttal.

---

> > > ### Author Response · Authors · 2024-08-09
> > >
> > > Dear reviewer qAvq, we are glad that we were able to resolve your key concerns and appreciate you raising your score. We will ensure that a more thorough discussion on the single vs. multi-threshold discussion along the lines of the rebuttal text is included in the camera-ready version, and thank you for pointing out this improvement. In case any additional questions arise during the discussion period, we would be happy to address them.

---

### Author Rebuttal · Authors · 2024-08-06

We thank all reviewers for their efforts, time and comments, which are greatly appreciated.

We are glad that you found the work **tackles an interesting and important problem** (qAvq: “very interesting and timely problem”; c7rs: “addresses an important problem”; LhAs: “interesting and important challenge”), proposes a **novel and well-grounded approach** (c7rs: “propose a novel approach”; LhAs: “framework is well-developed”, “integration of risk control into EENNs is novel”; q2CX: “theoretical analysis seems reasonable”) and appreciate the **extensive experimental validation** (qAvq: "experimentally demonstrated on a wide range of predictive ... and generative tasks"; c7rs: “empirical validation … on a range of vision and language tasks”; LhAs: "extensive experimental evidence across multiple domains"; q2CX: "The experiments on different vision tasks can be strong evidence to show the effectiveness…”). Reviewers also appreciated the method's post-hoc nature (qAvq: “does not add any computational overhead”), uncertainty aspect (c7rs: “consider both prediction quality and uncertainty estimation”), and the paper's writing style (qAvq: ”The paper is well-written … and a nice formulation.”).

We next address and clarify a key point raised in the reviews.

 **(1) Reliance on a single exit threshold (qAvq, q2CX)**

We agree with the reviewers that our reliance on a one-dimensional shared threshold among exit layers can be considered a limitation, as we highlight ourselves in §6. However, we think that **addressing the simpler, single-threshold case first provides a necessary stepping stone for extending our approach** to scenarios with higher-dimensional thresholds for EENNs. Note that for higher-dimensional cases new challenges arise both in terms of theoretical aspects (e.g., defining monotonicity requirements) and practical ones (e.g., substantially larger search spaces). As we highlight in §6, combining our work with some of the recently proposed risk control extensions for high-dimensional thresholds [1] could help address those challenges, and we think this is a very promising future direction.

Morever, we note that **single exit thresholds are quite common in the early-exit literature** [2, 3, 4, 5, 6, 7, 8, 9, 10], most likely due to the complications of designing principled threshold selection mechanisms for higher-dimensional cases. For instance, in [11] they resort to an (inefficient) heuristic based on evaluating randomly sampled threshold vectors on hold-out data. While we agree that the use of exit-specific thresholds might lead to even faster early-exit models, our work provides a principled selection mechanism for one-dimensional thresholds first, with hopes of extending to more complex higher dimensions in the future. Also, we are not aware of any work that studies in detail the potential inefficiencies induced by relying on a single threshold vs. multiple ones in early-exit architectures and we agree that this is an interesting an important direction for future work.

Lastly, as a workaround, one could **reduce the multi-dimensional problem to a single threshold** selection by defining the so-called *threshold function*. As an example, we would point to our language modeling experiment (§5.3 and L1120-1122 in Appendix C.3) based on the CALM model [2]. While in CALM the threshold is the same across exits for a given token, it changes between different tokens in a sequence (concretely, the threshold function is given by $f(\lambda,t) = \text{clip}_{[0,1]}(0.9 \cdot \lambda + 0.1 \cdot e^{-\tau \cdot t / N})$, where $\tau, N$ are fixed values, $t$ is the token index and $\lambda$ is the parameter to tune via risk control). Note that our current framework still supports such a dynamic threshold by performing risk control not on the threshold itself but on the parameter $\lambda$ of the threshold function. Similar ideas could also be used to support dynamic thresholds not only across tokens but also across layers/exits, allowing for the flexibility of exit-specific thresholds while keeping the risk control computations tractable.

We thank reviewers for pointing out this important aspect, and will add these details to the camera-ready version to better clarify our threshold assumption.

We further address each reviewers' concerns and questions in our individual responses.

**References**

[1] Teneggi, J., et al. How to trust your diffusion model: A convex optimization approach to conformal risk control. *ICML 2023*

[2] Schuster, T., et al. Confident adaptive language modeling. *NeurIPS 2022*

[3] Tang, S., et al. Deediff: Dynamic uncertainty-aware early exiting for accelerating diffusion model generation. *arXiv preprint 2023*

[4] Liu, Z., et al. Anytime dense prediction with confidence adaptivity. *ICLR 2022*

[5] Kaya, Y., et al. Shallow-deep networks: Understanding and mitigating network overthinking. *ICML 2019*

[6] Wołczyk, M., et al. Zero time waste: Recycling predictions in early exit neural networks. *NeurIPS 2022*

[7] Zhou, W., et al. Bert loses patience: Fast and robust inference with early exit. *NeurIPS 2020*

[8] Schwartz, R., et al. The right tool for the job: Matching model and instance complexities. *ACL 2020*

[9] Xin, J., et al. DeeBERT: Dynamic early exiting for accelerating BERT inference. *ACL 2020*

[10] Xin, J., et al. BERxiT: Early exiting for BERT with better fine-tuning and extension to regression. *EACL 2021*

[11] Elbayad, M., et al. Depth-adaptive transformer. *ICLR 2020*

---

### Decision · Program_Chairs · 2024-09-25

**Decision:**

Accept (poster)

**Comment:**

### Summary
This paper tries to improve the efficiency of deep models by exploring a type of Early-Exit Neural Network (EENN). EENNs accelerate inference by allowing intermediate layers to exit and produce a prediction early. However, this paper improves them by using approaches inspired by leveraging a method called as Risk Control (RC.)  RC manually sets the confidence thresholds for early exiting. The paper presents promising experimental results on vision and language tasks.

### Decision

The paper is well-written and clear. The approach is interesting, and the problem is important and potentially impactful. There were several criticisms directed toward the paper during the rebuttal period. However, the authors did a good job replying to them during the rebuttal period. The authors provided clarifications for many of the questions raised by the reviewers, and the proposed approach has merit on its own, as it studies a very interesting and often overlooked aspect of EE models and offers a holistic approach that demonstrates its effectiveness across models and tasks. Thus, I would recommend this paper for acceptance.

However, I recommend that the authors to read the rebuttals carefully and address the reviewers' concerns in the camera-ready of the paper.